# AQPX-cluster aquaporins and aquaglyceroporins are asymmetrically distributed in trypanosomes

Fiorella Carla Tesan [1,2], Ramiro Lorenzo [3], Karina Alleva [1,2,4✉] & Ana Romina Fox [3,4✉]

Major Intrinsic Proteins (MIPs) are membrane channels that permeate water and other small solutes. Some trypanosomatid MIPs mediate the uptake of antiparasitic compounds, placing them as potential drug targets. However, a thorough study of the diversity of these channels is still missing. Here we place trypanosomatid channels in the sequence-function space of the large MIP superfamily through a sequence similarity network. This analysis exposes that trypanosomatid aquaporins integrate a distant cluster from the currently defined MIP families, here named aquaporin X (AQPX). Our phylogenetic analyses reveal that trypanosomatid MIPs distribute exclusively between aquaglyceroporin (GLP) and AQPX, being the AQPX family expanded in the Metakinetoplastina common ancestor before the origin of the parasitic order Trypanosomatida. Synteny analysis shows how African trypanosomes specifically lost *AQPXs*, whereas American trypanosomes specifically lost *GLPs*. AQPXs diverge from already described MIPs on crucial residues. Together, our results expose the diversity of trypanosomatid MIPs and will aid further functional, structural, and physiological research needed to face the potentiality of the AQPXs as gateways for trypanocidal drugs.

[1] Universidad de Buenos Aires, Facultad de Farmacia y Bioquímica, Departamento de Fisicomatemática, Cátedra de Física, Buenos Aires, Argentina. [2] CONICET-Universidad de Buenos Aires, Instituto de Química y Fisicoquímica Biológicas (IQUIFIB), Buenos Aires, Argentina. [3] Laboratorio de Farmacología, Centro de Investigación Veterinaria de Tandil (CIVETAN), (CONICET-CICPBA-UNCPBA) Facultad de Ciencias Veterinarias, Universidad Nacional del Centro de la Provincia de Buenos Aires, Tandil, Argentina. [4] These authors contributed equally: Karina Alleva, Ana Romina Fox. ✉email: kalleva@ffyb.uba.ar; rfox@vet.unicen.edu.ar

The Trypanosomatida order of kinetoplastids (Euglenozoa, Discoba) gathers a vast diversity of parasitic protozoans that cause worldwide health problems infecting humans and livestock[1–3]. Drugs preferential uptake and the presence of pathogen-specific enzymes determine the selectivity and toxicity of currently available drugs for disease control[4,5]. In this regard, Major Intrinsic proteins (MIP) mediate the internalization of drugs that are the first choice against *Trypanosoma brucei* and *Leishmania* spp. (i.e., pentamidine and antimonial compounds, respectively)[6,7]. Those findings support MIPs as potential drug targets against protozoan parasites[8]. Regardless, channels with very different pore properties build up the MIP superfamily, and comprehensive analysis of trypanosomatid MIPs diversity is still missing.

MIPs facilitate the diffusion of water and a variety of relatively small solutes through biological membranes[9]. Even with considerable sequence divergence, inside the MIP superfamily, its members preserve a typical three-dimensional structure, and they organize as tetramers having each monomer an individual transporting pore[10]. Two NPA (Asn-Pro-Ala) motifs in the middle part of the pore, regulate water conductance and operate as a barrier for the passage of inorganic cations (such as $Na^+$ and $K^+$)[11,12], and also participate in proton filtration[13,14]. Still, protons are fully blocked at the selectivity filter[11,12,15], known as aromatic/Arginine (ar/R), which also executes a primary permeation role. The residues of this filter are related to the functional properties of the channel[16,17] and, interestingly, play a central role in trypanosomatid drug uptake, i.e., their mutation may lead to drug resistance events[18]. Finally, five amino acid residues, designated as Froger positions, are involved in the discrimination between water or glycerol transport[19].

The pioneer studies on MIPs diversity proposed that Eukarya isoforms derived from two bacterial channels: glycerol facilitators or aquaglyceroporins (GLP) and water channels or aquaporins (AQP)[20,21]. Subsequent studies revealed an unexpected diversity of MIPs in the three domains of life and that first distinction AQP versus GLP remained insufficient to describe MIPs phylogeny. Consistently, the nomenclature of MIPs became more complex. Today, four clusters of prokaryotic MIPs have been described, named as grades to point to the polyphyletic nature of the superfamily (AqpM, AqpN, AqpZ, and GlpF)[22]. In Eukarya, there are up to seven recognized families of land plant MIPs: plasma membrane intrinsic protein (PIP), tonoplast intrinsic protein (TIP), Nodulin 26-like intrinsic protein (NIP), small basic intrinsic protein (SIP), X or uncharacterized intrinsic protein (XIP), hybrid intrinsic protein (HIP), and GlpF-like intrinsic protein (GIP)[23–26], while green algae have PIPs and GIPs but also other five subfamilies (named MIP A–E) not found in land plants[24]. Animalia has four MIP families (AQP1-like, AQP8-like, AQP3-like, and AQP11-like)[26,27]. Phylogenetic studies including plants and animals cluster PIPs with AQP1-like (considered the classical AQPs), TIPs with animal Aqp8-like, and SIPs with AQP11-like[27,28]. There are different hypotheses regarding the origin of NIPs and AQP3-likes[27–29], which is still an unresolved issue. Nevertheless, there is currently no disagreement about the existence of a common ancestor among Eukarya AQP3-likes and Bacteria GlpFs so, the term GLP refers to this monophyletic group. On the other hand, the term AQP, when used, refers to a polyphyletic group.

As it is noticeable from the previous paragraphs, most of the described MIPs belong to two Eukarya supergroups (Archaeplastida and Amorphea -specifically Animalia-). In contrast, little is reported regarding other supergroups, such as Discoba, TSAR (Telonemia, Stramenopila, Alveolata, and Rhizaria), and Haptista. Still, the available data points to a quite diversified scenario in these supergroups. Within the TSAR supergroup, some MIPs cluster with the families PIP, GIP, and MIPE, whereas other MIPs cluster in a new family specific to TSAR organisms, named Large Intrinsic Proteins (LIPs)[30]. Also, there is no uniformity concerning MIP diversity among protozoans[28], while *Plasmodium* spp. (TSAR) carry a single *MIP* gene, up to five have been identified in the genomes of *T. brucei*, *T. cruzi*, and *L. major* (kinetoplastids, Discoba)[31]. *T. brucei* MIPs were previously set as GLPs and *T. cruzi* MIPs as AQPs, whereas *L. major* MIPs were described in both groups[28,32]. Additionally, *L. major* and *T. cruzi* AQPs were regarded as TIP-related AQPs[31,33]. However, none of those studies focused specifically on the phylogeny of the Kinetoplastea class MIPs. Today, the increased availability of genomes and transcriptomes of kinetoplastid species[34] provides the tools needed for a deep evolutionary study of MIPs diversity in this class.

Studies elucidating phylogenetic relationships among MIPs have opened ways to understand and predict relevant structure–function relationships in the evolution of utterly different organism lineages, such as tetrapods[22], insects[35], and plants[27]. In this work, we show that two MIP families expanded among trypanosomatids: GLP and a MIP family previously undescribed as such, named here AQPX. GLPs were not found in other kinetoplastid orders, whereas AQPXs were found in early-branching kinetoplastids. The AQPX family expanded in the Metakinetoplastina common ancestor before the origin of the parasitic order Trypanosomatida and extant trypanosomes hold up to four AQPX paralogs. Additionally, MIPs distribute asymmetrically inside the genus *Trypanosoma*: African trypanosomes specifically lost AQPXs and kept GLPs, whereas American trypanosomes specifically lost GLPs. This in-depth analysis of parasite MIPs may help understand the relevance of these channels in the physiology of the different parasites and assess their potential as drug targets.

## Results and discussion

**Kinetoplastid MIPs are either GLPs or non-orthodox AQPs.** We built a sequence similarity network (SSN) to explore where and how kinetoplastid MIPs localize in the superfamily sequence-function space. The starting point was a group of 52,453 MIPs retrieved from the Uniprot database. After clustering to 85% amino acid sequence identity and filtering by length, 16,170 representative accessions composed the network's nodes. The threshold for connecting nodes was set in an alignment score of 35 (corresponding to 35–40% pairwise sequence similarity) and rendered 10 clusters (Fig. 1a). Nearly half of the SSN nodes are from bacteria and the other half from eukaryotes, pointing to an expansion and diversification of the MIP superfamily that is similar in magnitude in both domains of life (Fig. 1b).

Already characterized MIPs that belong to different phylogenetic groups and with different permeation properties cluster separately in our SSN. Holding 80% of the nodes, Cluster 1 has a domain contribution similar to the full network, and the other smaller clusters are almost specific to Bacteria or Eukarya (Fig. 1b). MIPs with more divergent primary amino acid sequences localize in smaller clusters. That is the case for the plant XIPs and SIPs, the metazoan AQP11-12 group, algae MIPs (cluster 3, 5, 6, and 9, respectively), and other still uncharacterized divergent clusters (2, 4, 7, 8, and 10) (Fig. 1a). Figure 2 displays a detailed view of Cluster 1. Three main subclusters compose this cluster: (i) AQP$_{SSN}$ (also internally structured allowing us to distinguish plant PIPs, TIPs, and NIPs, metazoans AQP1-likes and AQP8-likes and, prokaryotic AqpZs, AqpNs and AqpMs); (ii) GLP$_{SSN}$ (where *T. brucei* and *T. evansi* MIPs localize among the Eukarya nodes), and (iii) AQPX$_{SSN}$ (a small subcluster of mostly uncharacterized MIPs). The subindex SSN highlights that

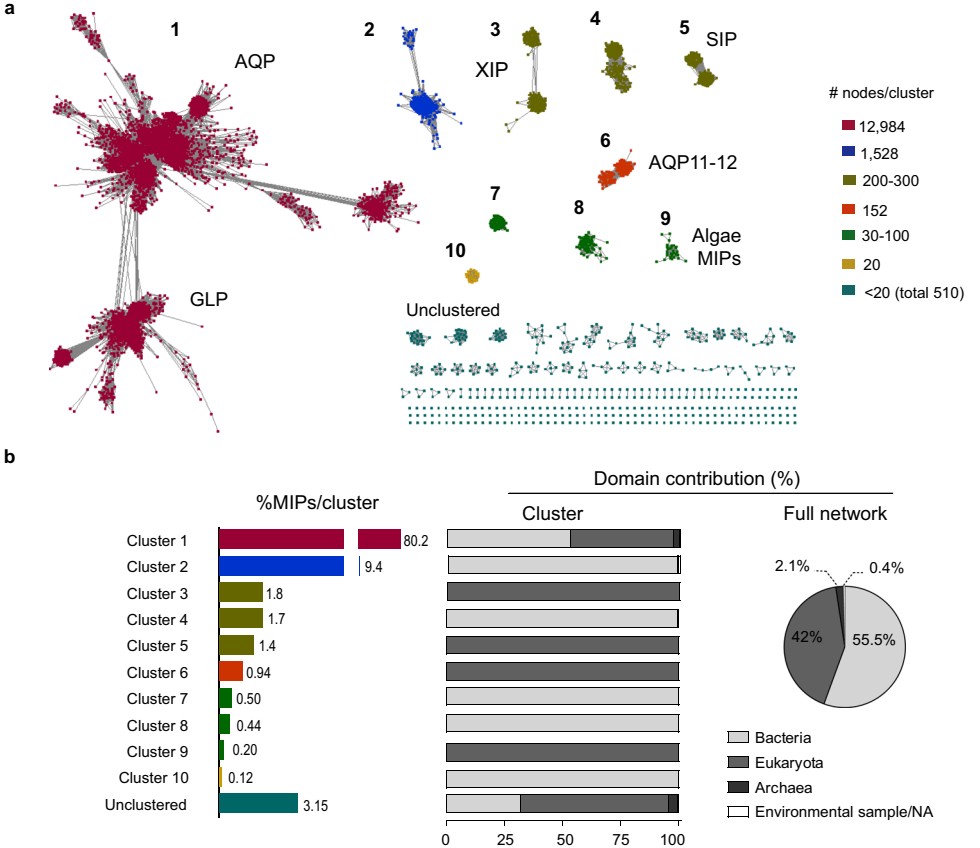

**Fig. 1 Sequence similarity network (SSN) of the MIP superfamily. a** SSN is composed of 16,170 nodes (squares), which represent proteins sharing >85% sequence identity, connected by edges with an average pairwise alignment score of at least 35. Edge length is a measure of the relative dissimilarity of each pair of sequences. Nodes are colored by the size of each cluster (numbered 1–10). Nodes found in clusters with <20 nodes were arbitrarily considered to uncluster. Names indicate clusters containing already characterized proteins. **b** MIPs distribution among clusters and taxonomic composition of each cluster according to the three-domain system of classification (Archaea, Bacteria, and Eukarya).

these groups arise from the network analysis and do not imply phylogenetic relations, even if both analyses can be congruent. Interestingly, many kinetoplastid (Discoba supergroup) MIPs are part of AQPX$_{SSN}$, a still uncharacterized subcluster that is far away from well-known MIPs.

**Kinetoplastid MIPs are abundant among AQPX$_{SSN}$ and AQPX is a MIP family**. The AQPX$_{SSN}$ subcluster is less crowded than the other two subclusters. Only 3% of Cluster 1 nodes are in this group. Long edges connect AQPX$_{SSN}$ with AQP$_{SSN}$, whereas none edges connect it to the GLP$_{SSN}$ (Figs. 1 and 2). AQPX$_{SSN}$ is composed of uncharacterized prokaryotic and eukaryotic MIPs. Almost 75 and 10% of the AQPX$_{SSN}$ nodes are from the Bacteria and Archaea domain of life, respectively (Fig. 2). The kineto-plastid MIPs, present in AQPX$_{SSN}$, have a unique closeness to prokaryotic uncharacterized MIPs. Thus, to investigate the putative origin of the MIPs that belong to the AQPX$_{SSN}$ sub-cluster, we performed a phylogenetic analysis of the prokaryotic MIPs. The study included those bacterial MIPs present in AQPX$_{SSN}$ (named AqpX) and the currently described four pro-karyotic MIP grades (i.e., AqpM, AqpN, AqpZ, and Glp)[22]. Supplementary Data 1 details sequence data. Our study showed that AqpXs integrate a well-supported grade among prokaryotic MIPs. Therefore, this is evidence of AQPX being a grade of MIPs whose origin can be placed before the emergence of the Eukarya domain of life. Detailed analysis and discussion of this task are available in the Supplementary Results and Discussion, in Sup-plementary Fig. 1 and 2.

Regarding the eukaryotic MIPs present in AQPX$_{SSN}$, all nodes belong exclusively to unicellular organisms, 72% corresponding to the Kinetoplastea class (Discoba supergroup) and 16% to the TSAR supergroup (Fig. 2). The SSN exposed that Discoba and TSAR supergroups have different MIPs distribution as already suggested[28]. Discoba MIPs distribute principally among the GLP$_{SSN}$ and AQPX$_{SSN}$ subclusters (41 and 57%, respectively), whereas TSAR MIPs are mainly from the GLP$_{SSN}$ subcluster (93%) with a low percentage of isoforms distributed among the AQP$_{SSN}$ and AQPX$_{SSN}$ subclusters (5 and 2%, respectively). Altogether, this data points to an important presence of AQPX isoforms inside the Discoba supergroup and not in other Eukarya supergroups.

**Asymmetric distribution of MIP repertoire among kineto-plastids**. It has been previously described that *T. cruzi* and *T. brucei* do not share any MIP ortholog, whereas parasites of the genus *Leishmania* share MIP orthologs with the former two[28]. Here, our SSN data stands out for the presence of AQPXs among trypanosomatids. To put all these data together and propose a hypothesis for the origin/s of trypanosomatid MIPs, we recon-structed MIPs phylogenetic history for the full Kinetoplastea class. We performed an intensive search of MIPs in publicly available databases and stumbled upon heterogeneous genome sequence availability (detailed in Supplementary Data 2). Try-panosomatida is the most studied order within the Kinetoplastea class with many genomes available, whereas the Prokinetoplastina subclass (*Ichthyobodo*, *Perkinsela*, PhM-4, and PhF-6) or

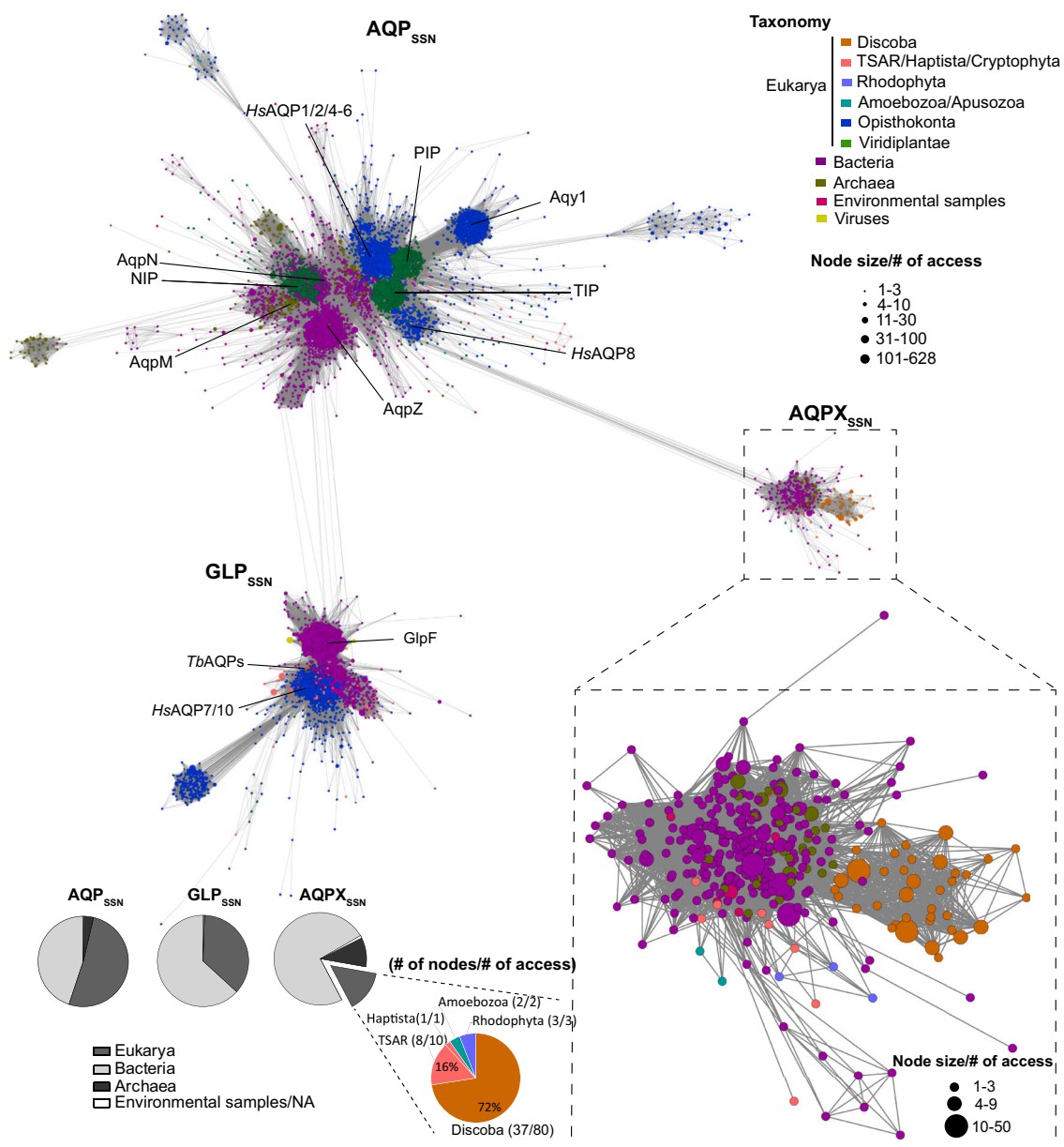

**Fig. 2 Close up view of Cluster 1 of the Sequence similarity network (SSN).** The cluster is composed of three main subclusters: AQP$_{SSN}$, GLP$_{SSN}$, and AQPX$_{SSN}$. Kinetoplastid AQPs localize in the AQPX subcluster. Nodes are colored to distinguish among taxonomic groups. The node size reflects the number of accessions that are grouped with 85% identity in that node. The pie charts show the taxonomic composition of each subcluster according to the three-domain system.

bodonida order have far fewer sequences available. Thus, we included transcriptome retrieved sequences to increase our data set of MIPs. In the specific case of *Parabodo caudatus* and *Procryptobia sorokini*, we retrieved their MIP sequences from studies where the bodonids were prey (Supplementary Data 3). We found no *MIP* sequences encoded in the genomes of two early-branching parasites/commensals (*Perkinsela* sp. and *Trypanoplasma borrelli*). Parasitism/commensalism evolved several times independently among kinetoplastids[36] (Fig. 3a) and, it seems that there is no relationship between this process and the *MIP* presence or absence in kinetoplastid genomes since, in opposition to *Perkinsela* sp. and *T. borrelli*, trypanosomatid parasites had many MIPs. Besides, the absence of *MIP* genes in a eukaryotic organism is a rare event that was only reported in three other protozoans: *Cryptosporidium parvum* (TSAR)[31], *Tetrahymena thermophila* (TSAR), and *Giardia intestinalis* (Metamonada)[28]. We also

searched for MIPs on species commonly used as outgroups in phylogenetic studies of kinetoplastids (i.e., euglenids or diplonemids). The complete list of MIPs here analyzed is reported in Supplementary Data 4. Curiously, the sequence identity among kinetoplastid MIPs and diplonemid or euglenid MIPs is low (Supplementary Data 5). Therefore, we searched for MIP sequences within the complete Discoba supergroup (which includes Jakobids, Heterolobosea, and Euglenozoa) to observe the big picture by constructing a preliminary phylogenetic tree. This tree, which also included bacterial MIPs as reference for each already described grade, was built by the Maximum likelihood method, and was rooted in the long and fully supported branch that separated GLPs from other MIPs (Fig. 3b). Thus, our tree displays two primary branches at first sight, generally referred to as GLP and AQP (Fig. 3b). Notwithstanding this central division, we acknowledge the polyphyletic nature of the AQPs, further

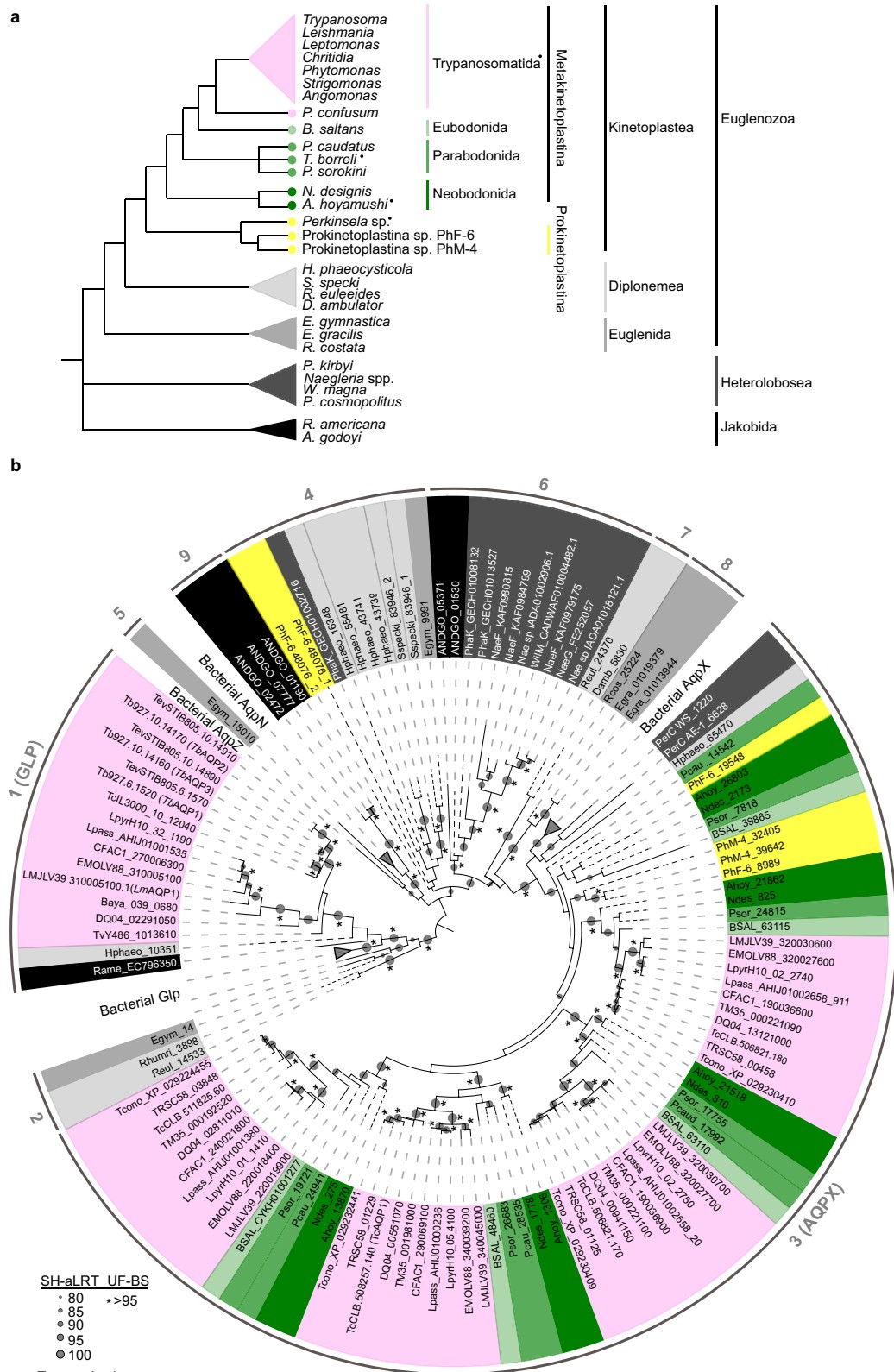

**Fig. 3 Preliminary phylogenetic tree of the Discoba Major intrinsic proteins (MIPs) superfamily. a** Cladogram showing the current accepted evolutionary relations inside the Discoba supergroup[36,77,78]. Genera and species present in this cladogram correspond to those with genome/transcriptome data analyzed in the current work. Parasitic species or genera are indicated by full black circles superscripts. **b** Discoba MIPs preliminary phylogenetic tree reconstructed by maximum likelihood. Branch support was assessed by the ultrafast bootstrap (UF) approximation with 10,000 replicates and the SH-aLRT with 1000 replicates. Accessions were shaded following the same color code of the cladogram in **a**. Dashed lines represent transcriptome retrieved sequences and each MIP clade is referenced by a number.

explained over the text by describing each AQP group found (referenced by consecutive numbers, 3–9, in Fig. 3b) and focusing later precisely on AQPXs.

Inside the Euglenida group, only one phototrophic organism (the freshwater *Euglena gracilis*) has a sequenced genome and transcriptome available, and other two organisms (the phototrophic *Eutreptiella gymnastica* and the heterotrophic *Rhabdomonas constata* have transcriptomes available. *E. gymnastica* has two AQPs (Fig. 3b, groups 4 and 5) that do not group with the other euglenid isoforms (Fig. 3b, group 8). Besides, *R. costata* and *E. gracilis* AQPs, located in a long branch of the tree, have unique MIP structural determinants (Supplementary Data 6) and low sequence identity to all the other Discoba MIPs (under the 20%) (Supplementary Data 5). Three species are not enough to build up conclusions about the entire group, but it allows us to expose that lineage-specific MIPs evolved among euglenids and, none of them are ancestors of kinetoplastid MIPs. *Andalucia godoyi* (Jakobids) is the unique organism that we found to have MIPs grouping with AqpNs (Fig. 3b, group 9). Also, AQPs from *A. godoyi* and heterolobosean species integrate a supported group with low amino acid sequence identity to the other Discoba MIPs (Fig. 3b, group 6) and with >40% sequence identity to plant TIPs (NCBI BLAST results, 65–70% coverage). Group 4, even without significant statistical support, clusters Heterolobosea and Euglenozoa MIPs, keeping structural determinants that resemble AQP1-like channels or plant PIPs (Supplementary Data 6, selectivity filter), both proposed to derive from a common eukaryotic ancestor[27]. Interestingly, just AQPs from Prokinetoplastina and none of the trypanosomatid MIPs form part of those previously described Discoba AQP groups (4–9). Instead, all trypanosomatid MIPs from the AQP branch belong to the AQPX cluster with Bacteria AqpXs (Fig. 3b, group 3). There exists the possibility that a trypanosomatid ancestor acquired an *AQPX* by lateral gene transfer, an event already described for several trypanosomatid genes[37]. But the *AQPXs* were present in early-branching kinetoplastids (Prokinetoplastina), and therefore are ancestral kinetoplastid genes. Thus, if the acquisition of *AQPXs* occurred by lateral gene transfer, it happened before the kinetoplastid lineage emerged.

In opposition to the vast number of AQPXs, our analysis revealed a small number of GLPs among kinetoplastids (Fig. 3b, group 1). Moreover, we found only trypanosomatid GLP isoforms, and we found no bodonid, nor prokinetoplastina GLPs, suggesting an asymmetric MIPs repertoire among kinetoplastids. Without considering trypanosomatids, we found only five GLPs. One diplonemid (*Hemistasia phaeocysticola*) and one jakobid (*Reclinomonas americana*) isoform showed 19–29% identity to trypanosomatids GLPs and similar structural determinants (Supplementary Data 5 and Supplementary Data 6). While other two GLPs from diplonemids (*Rhynchopus* spp.) and one from a euglenid (*E. gymnastica*) (Fig. 3b, group 2) had lower sequence identity to trypanosomatids GLPs (15–24%) and different structural determinants (Supplementary Data 5 and Supplementary Data 6). Comparing among the trypanosomatid species, we observed that African trypanosomes (*T. evansi*, *T. congolense*, *T. vivax*, and *T. brucei*) have only GLP representatives and no AQPXs. Also, outside the Trypanosoma genus, the genome of *Blechomonas ayalai* codified only for a GLP. On the contrary, American trypanosomes (*T. theileri*, *T. rangeli*, *T. conorhini*, and *T. cruzi*) have four MIPs, all of which are AQPXs, and none GLP. *T. grayi* remains an exception to this matter as its genome codes for the four AQPXs and one GLP, similar to the genomes of *Leishmania* spp.

Finally, to evaluate the reliability of the heterogeneous sources of Discoba MIPs we analyzed the completeness of the genome and transcriptome assemblies using the tool Benchmarking

Universal Single-Copy Orthologs (BUSCO). Most of them showed good levels of completeness (Supplementary Data 2, analyzed in Supplementary Results and Discussion). Additionally, most of the transcriptomes here analyzed were already used to successfully carry out a comparative analysis of euglenozoans metabolic enzymes and molecular features (DNA pre-replication complex, kinetochore machinery)[34]. Altogether, this indicates that a reliable set of assemblies was used in our MIPs searches. Still, it is worth mentioning that a different picture might be reconstructed once more Discoba organisms have their genomes sequenced and can be included in the study.

**Origin of the trypanosomatid AQPα-δ clades in the Metakinetoplastina group.** Our preliminary analysis showed that the GLP grade was less crowded than the AQP group, as if an expansion among AQP grades had occurred. This burst can be seen specifically in the AQPX family, populated by trypanosomatids. Thus, to better understand kinetoplastid AQPXs' evolutionary history, we built a phylogenetic tree for the Discoba supergroup analyzing a wider diversity of trypanosomatids. We added the early-branching trypanosomatid, *Paratrypanosoma confusum*, the plant infecting *Phytomonas*, and the monoxenous genera *Angomonas* and *Strigomonas*. The AQPX isoforms of the early-branching Discoba organism *Percolomonas cosmopolitus* (Heterolobosea) served as root.

In this tree, trypanosomatid AQPX isoforms segregate together with bodonid MIPs in four very well-supported orthologous clusters: α, β, γ, and δ (named after *T. cruzi* and *L. major* aquaporins[31]) (Fig. 4). Each cluster is internally congruent with the organismal tree at species levels and, within each one, sequence identities go from 50 to 90% (Supplementary Data 7). AQPXs from Prokinetoplastina, early-branching kinetoplastids, compose a sister clade of these α-δ clades. AQPXs of free-living bodonid (eu-, para-, and neo-bodonids), and the only diplonemid AQPX found, form a more distant clade from trypanosomatid AQPXs, but this node is not statistically supported (Fig. 4). Altogether, we propose that the α-δ loci appeared through gene duplication from a single ancestral locus in the genome of an ancestral metakinetoplastid before the diversification of extant genera.

**Gains and losses of MIPs in Trypanosome genomes.** Inside the Trypanosomatida order, the genomes are highly syntenic[38], even though our phylogenetic analysis showed important differences in the displayed MIPs repertoire exposed by its members. Thus, to get more clues about trypanosomatid MIPs history, we compared the genomic neighborhood of these channels among representative trypanosomatids and their closest known non-parasitic relative, *B. saltans* (Fig. 5). We analyzed nine genomes, four of them are assembled at the chromosome level (*T. cruzi*, *T. brucei*, *T. congolense*, and *L. major*), two at the supercontig level (*P. confusum* and *B. saltans*) and three at the contig level (*T. grayi*, *T. theileri* and *B. ayalai*) (Supplementary Table 1). Overall, the quality of the assemblies, even if not homogeneous, is undoubtedly good. The genome coverages for the studied regions are among 41X and 200X (Supplementary Data 8). The coverage and undefined regions (Ns) are available in Supplementary Figs. 3–8. In Fig. 6, we summarized the accumulated knowledge relative to Discoba MIPs diversity. Inside the Kinetoplastea class, we propose a scheme of gains and losses compatible with our phylogenetic and syntenic data (Fig. 6a).

There is conserved synteny for the *α-δ AQPXs* of trypanosomatids and *B. saltans* (Fig. 5a and Supplementary Figs. 3–5) even when this bodonid genome only showed ~10% co-linearity with trypanosomatid genomes[39]. A fifth *AQPX* in *B. saltans*, with low

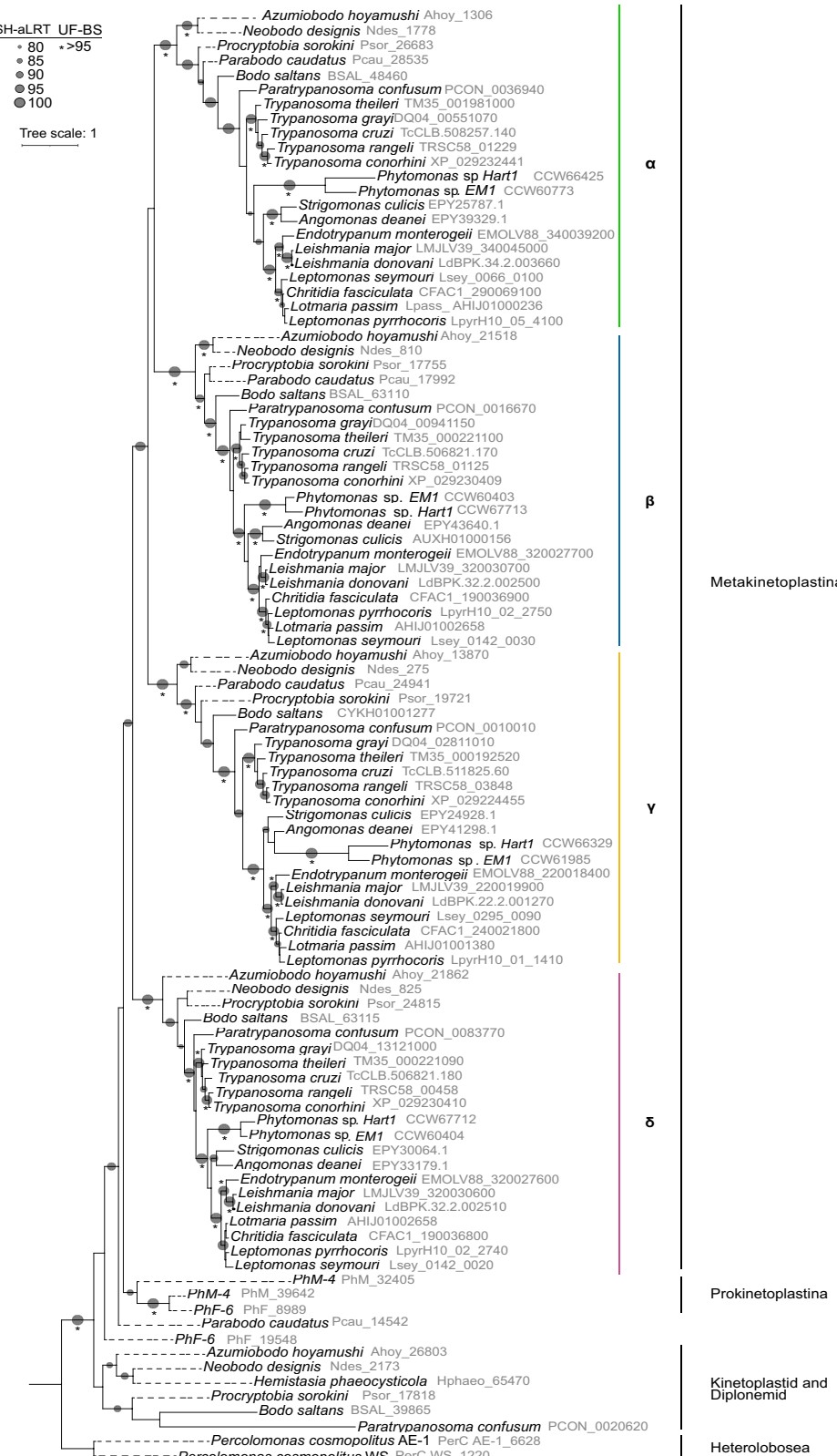

**Fig. 4 AQPX phylogenetic tree of the Discoba supergroup.** Phylogeny of AQPX proteins was reconstructed by maximum likelihood. Branch support was assessed by the ultrafast bootstrap (UF) approximation with 10,000 replicates and the SH-aLRT with 1000 replicates. Dashed lines represent transcriptome retrieved sequences.

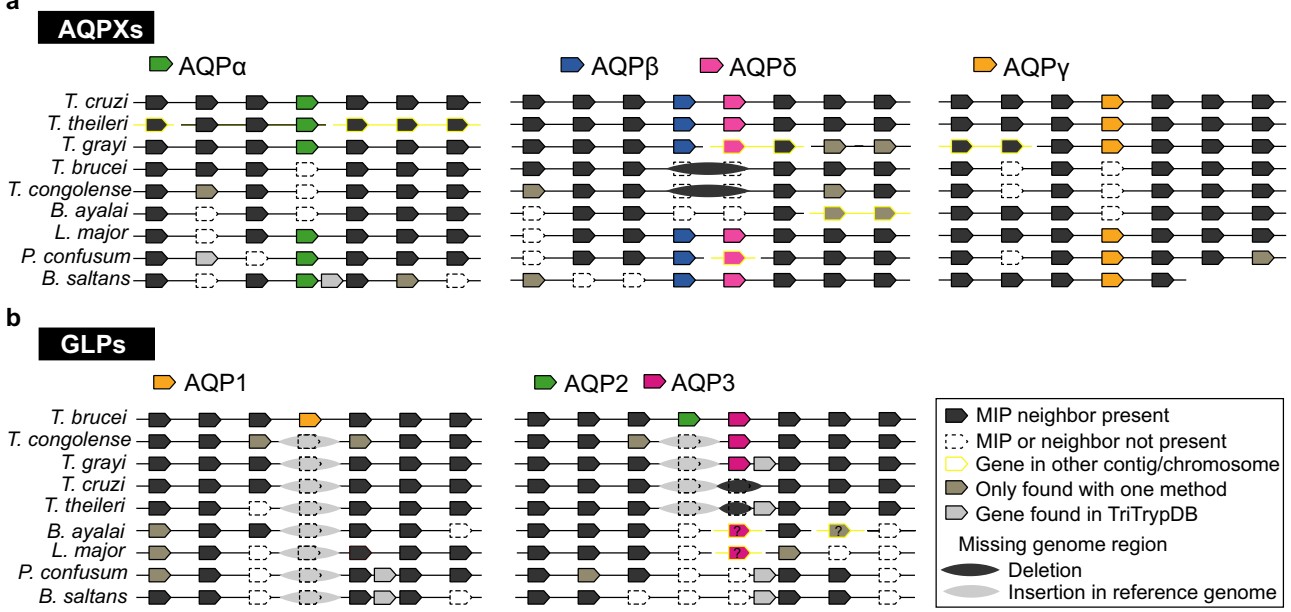

**Fig. 5 Synteny analysis of Trypanosomatid MIPs.** Comparative gene organization of the regions where **a** AQPXs and **b** GLPs are found. The analyzed regions comprise 10 Kb down and up-stream of each MIP in the reference genomes (T. cruzi or T. brucei) and the equivalent syntenic regions in the other genomes. For clarity, in this figure only the first three genes for down and up-stream regions are shown. Homologous genes are vertically aligned. The graph shows when the syntenic genes are observed using both SimpleSynteny analysis and TritrypDB (black arrow box) or one of these methods (dark gray arrow box). Genes absent in the reference genome (T. cruzi or T. brucei) were not detected by SimpleSynteny but were observed in TritrypDB (light gray arrow box).

sequence identity with all the other AQPXs, localizes in a genomic region non-syntenic with parasite genomes, neither with the fifth AQPX of P. confusum (Supplementary Fig. 6). Then, the genome region coding for this B. saltans AQPX probably was lost in the trypanosomatids ancestor during the genome rearrangement in the transition from free-living to parasitism.

Among trypanosomatids, α-δ AQPXs seem to have been lost two times in different branches of the evolutionary tree (in African trypanosomes and B. ayalai, Fig. 6a).

Even when the AQPXs are missing in these two groups, the flanking genes are conserved (Fig. 5a). In the particular case of α and γ AQPXs, the accumulation of mutations seems to be the mechanism of gene losses, as the size of the intergenic region among flanking genes is close to 1 Kb, the expected size for these AQPs, (Supplementary Figs. 3–5). The β and δ AQPXs localize in tandem in trypanosomatids and B. saltans (not in P. confusum), and different mechanisms seem to be after these gene losses. In African trypanosomes, β and δ AQPXs losses seem a consequence of a deletion in their most recent common ancestor genomic region. In contrast, their losses in B. ayalai seem not to be associated with genomic deletions but with the accumulation of mutations (Fig. 5a, and detailed synteny data in Supplementary Fig. 4).

The closely related species T. brucei and T. evansi are the only two trypanosomatids carrying three GLPs (Fig. 6b). TbAQP1 neighbor genes are conserved inside the Trypanosomatinae subfamily. In contrast, none of TbAQP1's orthologs appear in that syntenic region. Thus, TbAQP1 (and its ortholog in T. evansi) appears to be a recent acquisition, via transpositive duplication, in their last common ancestor (Fig. 5b). TbAQP2 and TbAQP3 localize in tandem in T. brucei chromosome 10 and, TbAQP2 seems to be a consequence of a recent duplication event as TbAQP3 has a higher sequence identity with the GLPs of the other trypanosomatids than TbAQP2 (T. congolense, T. grayi, L. major, and B. ayalai). The TbAQP2-3 genomic region is syntenic

within the subfamily Trypanosomatinae, missing the GLPs only in American trypanosomes (Fig. 5b). Nevertheless, synteny is not conserved in this region among the subfamilies Leishmaniiae and Trypanosomatinae. That is congruent with the analysis reported by El-Sayed et al.[38] of the syntenic blocks among L. major and T. brucei (neither TbAQP2-3 nor LmAQP1 genome regions are in the described syntenic blocks). Besides, no GLP genes were found in P. confusum or B. saltans genomes. The orthologs of TbAPQ3 flanking genes are retained but the intergenic region among those genes is large in B. saltans (near to 4 Kb) and even larger in P. confusum (near 35 Kb) (Supplementary Fig. 7). Moreover, this large region of P. confusum is undefined and therefore we cannot exclude the presence of a GLP in there. Therefore, we assembled transcriptomes available for P. confusum (Supplementary Data 2) and searched for GLPs, finding none. To complete the analysis, we also searched for GLPs in B. saltans transcriptomes (Supplementary Data 2), and we found none either. We can think that TbAQP3 orthologous genes were specifically lost in these species. But, outside trypanosomatids, kinetoplastids lack GLPs, and the scenario of GLP loss in every lineage is very improbable. The most parsimonious scenario is the acquisition of a GLP in the common ancestor of trypanosomatids (after P. confusum branched at the Trypanosomatidae family base) which was then lost precisely two times: in American trypanosomes and the subfamily Strigomonadinae (Fig. 6a, b).

So, genera and species-specific gene gains and losses resulted in an asymmetric repertoire of MIPs in extant trypanosomatid parasites. Such processes are usual in the evolutionary history of other protein families among T. brucei, T. cruzi, and Leishmania species (i.e. cathepsins, amastins, nucleoside, and amino acid transporters)[39,40]. Utterly different lifestyles and hosts might relate to species-specific gene expansions and losses. For example, amastin diversity remained unchanged until the origin of Leishmania. So, the specific δ-amastin expansion that occurred in this species was speculated to relate to Leishmania's vertebrate

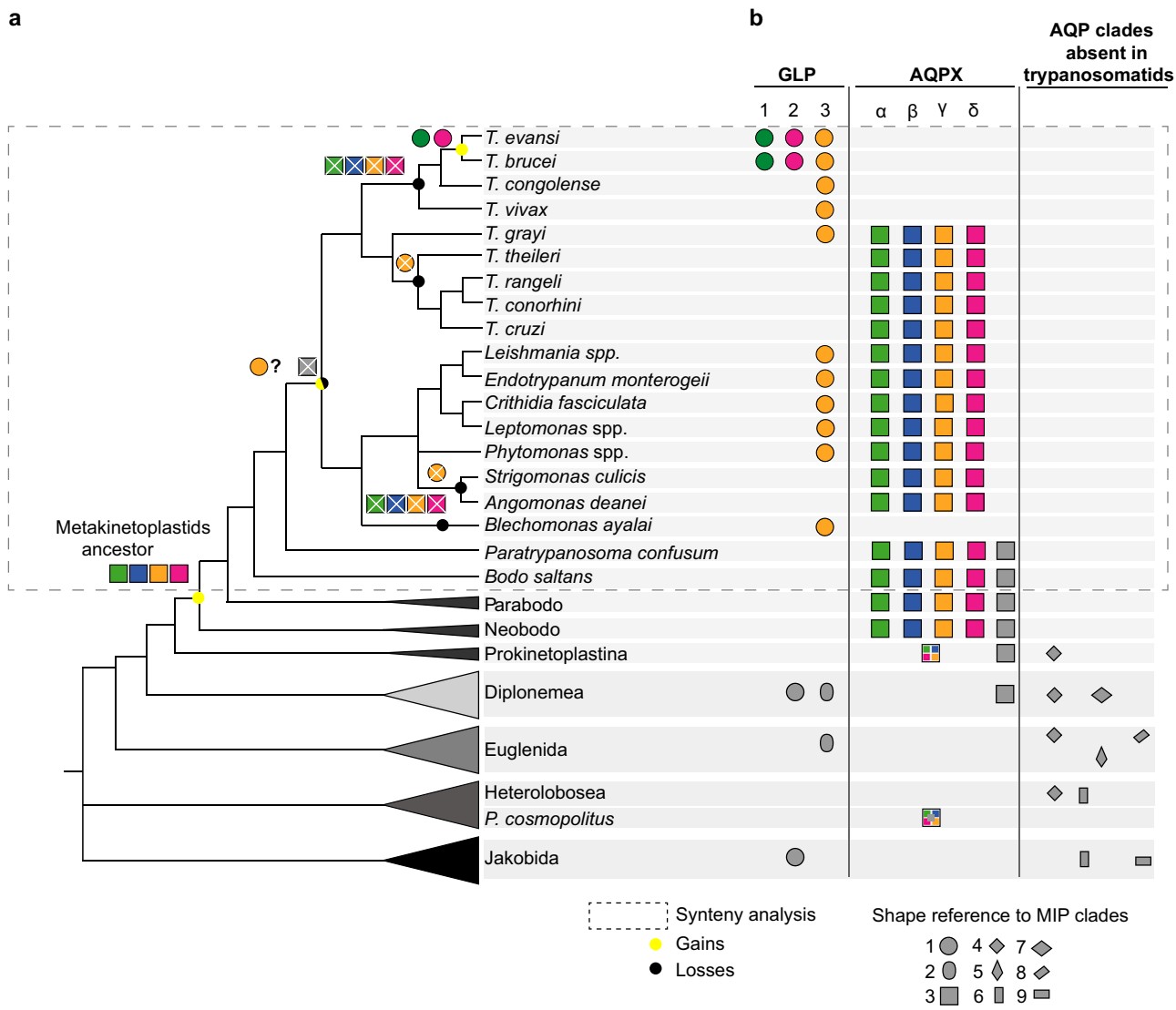

**Fig. 6 Proposed evolutionary history of Trypanosomatid MIPs. a** Cladogram showing the current accepted evolutionary relations inside the Discoba supergroup for the species analyzed in this work. **b** The MIPs repertoire for each species or group of species is represented by geometric shapes (proposed homologous genes share the same shape). Each shape is linked to a number in the references, and a number connects to the preliminary phylogenetic tree from Fig. 3. A dashed line square delimits the taxonomic groups represented by the syntenic analysis. Inside Metakinetoplastina, orthologs are represented in the same color. Inside the AQPX clade, for simplicity and because the phylogenetic relationships of some AQPXs are not fully resolved, gray squares may represent more than one homologous isoform. Multicolor squares represent ancestral homologs of trypanosomatid AQPXs. Over the cladogram, a proposed scheme of gains and losses, of GLPs and AQPXs, is depicted.

parasitism given the absence of this gene family in related monoxenous species (insect-restricted parasitism)[40]. Regarding the MIP superfamily, biological relevance of each family (GLP and AQPX) in trypanosomes still remains obscure though the asymmetric pattern is coherent with the proposal of an evolutionary relationship between the loss of *AQPs* and consequent expansion of *GLPs* (or the other way around) based on observations of other unicellular organisms like Oomycetes, that hold numerous GLP isoforms and none AQPs[28].

**Key structure determinants of kinetoplastids AQPXs**. To gather evidence of the putative role of MIPs in the evolution of kinetoplastids, we analyzed those key residues known to be related to the function and selectivity of the channels (i.e., the two signature NPA motifs, the selectivity filter and the Froger positions).

When GLPs are analyzed, it emerges that most of the trypanosomatids hold the same amino acids in NPA, selectivity

filter, and Froger Positions (except the extremely variable P5) (Fig. 7a). Among these isoforms, some have been functionally characterized as permeable to several solutes (Supplementary Data 6). For example, *Lm*AQP1 facilitates the diffusion of water and many non-ionic solutes (methylglyoxal, glycerol, dihydroxyacetone, glyceraldehyde, erythritol, and adonitol) but not urea[41]. Also, this GLP acts as a metalloid (As and Sb) gateway with implications in therapeutic interventions[42].

The most recently acquired GLP of *T. brucei* and *T. evansi* (AQP2) present utterly divergent key MIP residues from the other GLPs (Fig. 7a). These AQP2s are the only GLPs with non-canonical NPA motifs (NSA and NPS). Importantly, the N in the first position of the motifs that have been proved to be important for cation blockage[11,12] is conserved in *T. brucei* and *T. evansi* AQP2. Interestingly, functional consequences of the absence of both classical NPA motifs in *Tb*AQP2 are related to pentamidine sensitivity since the restitution of the NPA-NPA blocked the uptake of the drug[7].

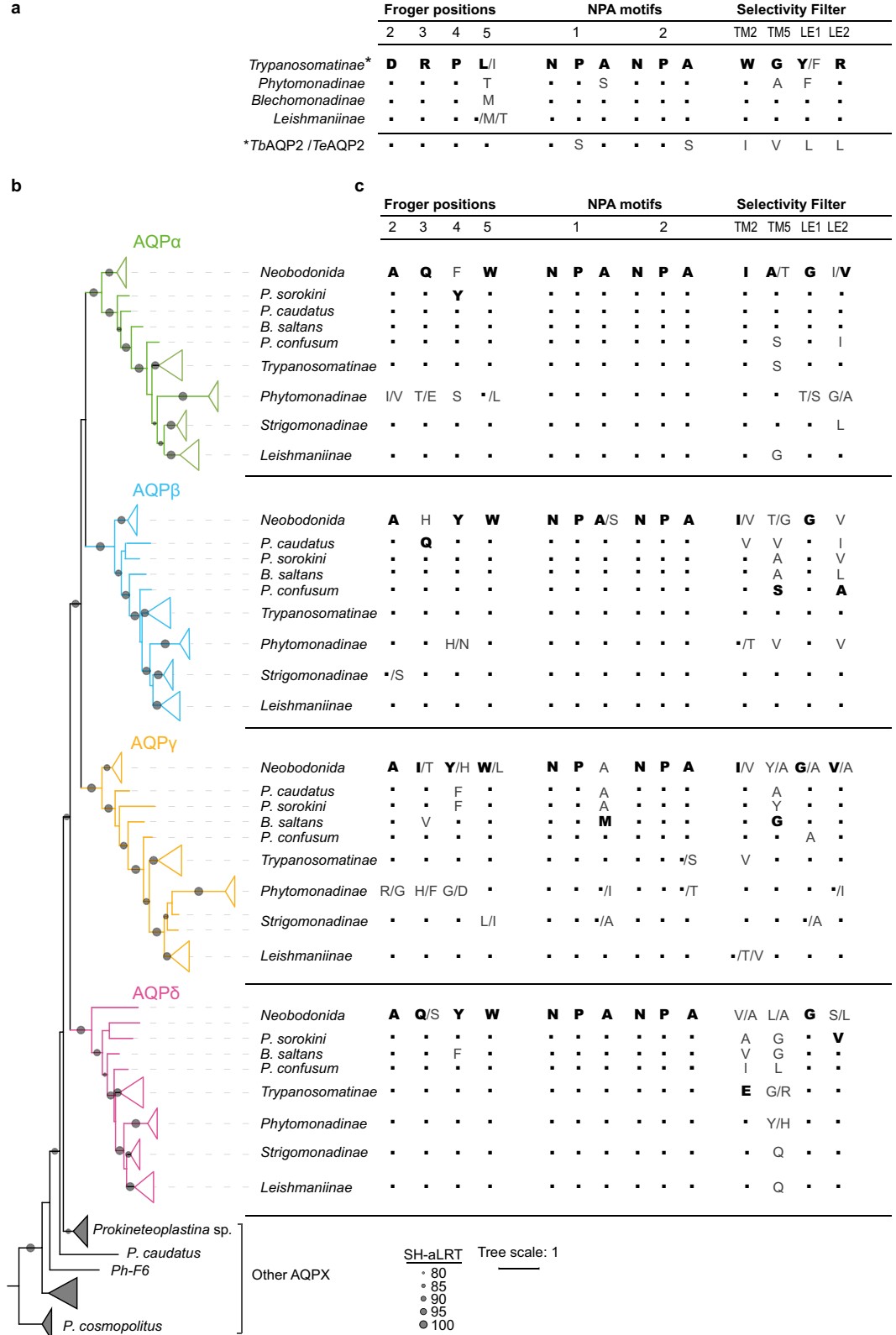

**Fig. 7 Key MIP residues from GLP and AQPX subfamilies.** Froger positions, NPA motifs and Selectivity filter residues of Discoba **a** GLPs and **b**, **c** AQPXs. The most frequent amino acid of a particular position in each clade is shown in bold first and then with dots. **b** Collapsed Discoba AQPXs phylogenetic tree reconstructed by maximum likelihood. **c** Analysis of residues.

Regarding the selectivity filter, these AQP2 carry a rare signature (IVLL), which is drastically different from the fully conserved selectivity filter of other trypanosomatids, or even Discoba GLPs (WGYR)[7,43] (Fig. 7a and Supplementary Data 6). *Tb*AQP2 selectivity filter is wider and more aliphatic than others. A first hypothesis sustains that this feature contributes to pentamidine passing through[7,44] and a second one that the unique selectivity filter in combination with a consequently exposed Asp (D265, Froger position P2), allows a high affinity binding of pentamidine followed by endocytosis[45]. It is vital to bear that no other *T. brucei* MIP participates in pentamidine uptake (*Tb*AQP1 nor *Tb*AQP3), whereas all *T. brucei* MIPs facilitate the diffusion of water, glycerol, and metalloids in a similar way[46,47]. *Tb*AQP2 also presents a very different expression pattern compared to *Tb*AQP1 (the most abundant MIP in *T. brucei*) and *Tb*AQP3 (only present in blood stages)[46]. Also, *Tb*AQP2 and *Tb*AQP3 have different subcellular localization and might play different roles in permeating water, glycerol and still undiscovered solutes[46,48,49]. *Tb*AQP2 changes in key residues plus the different localization and transcription levels among paralogs point to their neofunctionalization in the last common ancestor of *T. brucei and T. evansi*.

On a different note, AQPXs distribute among kinetoplastids. All orders (Tyrpanosomatida, Bodonida, and Prokinetoplastina) have at least two AQPXs. We analyzed key MIP residues, sorted by kinetoplastid orders and even subfamilies (Fig. 7b). AQPXs display generally conserved Froger positions (AQYW from P2 to P5) (Fig. 7c) with AQP-like residues occupying them. Regarding the NPA motifs, while α, β, and δ AQPXs present well-conserved NPAs, AQPγs present the first (N-terminal) motif as NPM (Fig. 7c and Supplementary Data 6). Interestingly, this substitution is absent in prokinetoplastina, para- and neo- bodonids, suggesting that it occurred in the common ancestor of *B. saltans* and trypanosomatids. Two isoforms from the γ clade carry neither classical NPA motifs: *Tc*AQPγ (NPM-NPS) and AQPγ from *Phytomonas* sp. EM1 (NPI-NPT) (Supplementary Data 6). Currently, there is no data regarding the permeation capabilities of these last two AQPγ, but there is some information about other members of the γ clade with an N-terminal NPM motif. Homology modeling studies suggest that *Lm*AQPγ maintains a well-conserved core structure[50], and functional studies showed that the *Ld*AQPγ is so far the only AQPX of this parasite that facilitates water permeation[32]. Confirmation of these results for other clade members could reveal a neofunctionalization of this AQPX in the last common ancestor among the free-living *B. saltans* and trypanosomatids.

AQPXs have a rare pattern that resembles none of the previously described selectivity filter for the different families of AQPs (FHTR, PIPs; FHCR, AQP1-like; HIA/GR/V, TIPs; HIA/GR, AQP8-like; and T/PL/VAL, unorthodox AQPs)[51]. Compared with selectivity filter in classical water channels AQP1-likes and PIPs, the selectivity filter of AQPXs do not keep the R in Loop E (LE2), nor the aromatic amino acids in TM2, having, instead, aliphatic residues (Fig. 7c). That may give place to more hydrophobic and broader filters. Though their selectivity filter is aliphatic, they also hold an aliphatic uncharged residue (an A) where *Tb*AQPs have an acidic amino acid (Froger position P2) and the impact of these differences and the eventual exposure of other AQPX residues affecting permeation or selectivity needs to be addressed by further structural and functional research. Finally, many AQPXs (except the β orthologs) have a V in the LE2 position. The presence of a V in this position was reported as a signature for subcellular MIPs[52]. Consistently, *Tc*AQPα is present in acidocalcisomes and a vacuolar structure near the flagellar pocket[53,54], *Ld*AQPα and δ in subcellular structures[32]. From a functional aspect, none of those mentioned MIPs

(*Tc*AQPXα, *Ld*AQPXα, *Ld*AQPγ, and *Ld*AQPδ) allow glycerol permeation[32,53]. This functional data was not expected because, as already mentioned, a wider selectivity filter seems to be present in AQPXs[50].

Recently proposed permeation mechanisms through *Tb*AQP2[7] allow us to ask whether AQPXs might be capable of facilitating the uptake of larger solutes. However, as mentioned above, they have so far poor water or glycerol permeability. They may present a different solute selectivity profile given their rare selectivity filter, they might have additional undescribed pore constrictions, or there might be still unknown regulatory factors stabilizing their open or closed states influencing heterologous expression results and conclusions. It is worth mentioning that conclusions based on MIP motifs and their respective consequences on pore sizes and selectivity profiles can only be reached on the bases of structural results. Crystallization or ab initio/homology combined models need to be pursued to elucidate Kinetoplastid AQPX structures given their low identity with already crystallized MIPs.

In conclusion, we depicted here the complex universe of MIPs through a SSN, clearly exposing that trypanosomatids carry GLPs and AQPXs. AQPXs compose a cluster far away from the already characterized MIPs and, our phylogenetic studies support that they integrate, to the best of our knowledge, a newly defined MIP family. We got an insight into the phylogenetic study of these channels in kinetoplastids. We found that the α-δ clades appear in the common ancestor of bodonids and trypanosomatids. Curiously, African trypanosomes lost all the AQPX isoforms. Instead, these trypanosomes hold GLPs that we proposed to be acquired in a trypanosomatid ancestor and specifically lost in American trypanosomes. Was this change of MIPs repertoire inside the Trypanosomatinae subfamily a gene replacement process among *GLPs* and *AQPXs*? AQPXs hold selectivity filter residues that allow us to speculate that they have a more hydrophobic and wider selectivity filter than classical AQPs. Then, can the solutes permeated by AQPXs possibly be similar to GLPs permeated ones? As already exposed, AQPX do not seem to have good glycerol permeability. Nonetheless, the nature of biologically relevant solutes that permeate these channels is still elusive. Future research on the permeation capability and structure of GLPs and AQPXs will help understand their importance in the parasite's physiology. That, together with the knowledge on MIP repertoire and evolutionary history are crucial steps to unveil possible drug sensitivity/resistance mechanisms in the treatment of trypanosomiasis.

## Methods

**Construction of sequence similarity network**. The SSN of the MIP superfamily was generated using the EFI-EST server[55]. The full-size Pfam PF00230 database was downloaded from UniProt (version 2020-02). Proteins were clustered at 85% amino acid sequence identity using h-cd-hit[56] and filtered by length (200–500 residues). A list of 16,170 accessions, representative of 52,453 sequences, was loaded in the EFI-EST server (Option D). An alignment score of 35 (corresponding to ~40% sequence identity) was used to generate the SSN. The resultant network of ~8 M edges was visualized in the open-source software Cytoscape 3.8[57] using a 64 GB RAM server (Supplementary Data 9).

**Sequence retrieval and phylogenetic analysis**. To build the prokaryotic MIP tree protein sequences already known to belong to specific MIP families (i.e., AqpM, AqpN, AqpZ, Glp) were retrieved from Pommerrenig et al. (2020)[58], and together with the prokaryotic AqpX sequences retrieved from our SSN analysis, were clustered at 60% amino acid sequence identity using h-cd-hit[56]. Sequences were aligned using MAFFT[59] v7 and trimmed using TrimAL[60] (-g 0.8 -cons 65). The list of accessions is in Supplementary Data 1.

Protein sequences from the Discoba supergroup organisms were retrieved from the public databases TriTrypDB, NCBI, and iMicrobe. First, we included MIP sequences that were tagged as aquaporin in the database, and we used a tBLASTn strategy to expand our set of MIPs. When no available genome was found for a given organism, we searched within transcriptome, either by blasting within published and publicly available assemblies or by assembling the Sequence Read Archive (SRA) using the rnaSPAdes software[61] in the Galaxy servers at usegalaxy.

org.au and usegalaxy.org[62]. The sequence assemblies from Butenko et al.[34] were provided by Dr. Lukeš lab. Parabodonida MIPs were retrieved from studies where *P. caudatus* and *P. sorokini* were prey. RNAseq from samples that contained the Parabodonida and other species (PhF-6, *Rhodelphis limneticus*, *Rhodelphis marinus*) and cleaned RNAseq from those species were compared. Sequences were considered as putative Parabodonida MIPs, analyzing their identity among different RNAseq (Supplementary Data 3) and observing their position in the phylogenetic tree. MIP sequences wrongly assigned to *Colpodella angusta* (NCBI) were confirmed to belong to its prey, *P. caudatus*. *C. angusta* supposed MIPs that were only partial were almost identical to the retrieved *P. caudatus* MIPs (sequence identity climbed to 98 and 99%). Additionally, no genomic nor transcriptomic data was found for *Ichtyobodo* (Prokinetoplastina), *Cryptobia* (Parabodonid, Metakinetoplastina), *Dimagistella* spp., *Klosteria*, *Rhynchobodo* sp., or *Actuariola* (Neobodonids, Metakinetoplastina). *Percolomonas cosmopolitus* cultures were fed with *Enterobacter aerogenes*. Thus, the presence of bacterial contaminating transcripts was tested for the strain WS. Megablast of *Percolomonas cosmopolitus* Strain WS assembly against BLAST nucleotide database (nt17-Apr-2014) showed that 524 of 11,058 query sequences had a match (cut off e-value $10^{-3}$). When selecting only the first match for the 524 query sequences (hit lowest e-value), 26 query sequences were matching with bacterial sequences. Suggesting very low contamination with bacterial RNA (26/11,058) and none of the matches correspond to the MIPs found in the transcriptome. The quality and completeness of the proteomes, transcriptomes, and genomes used in this study were assessed by using BUSCO tool suite v5.0.0[63] in the Galaxy public servers at usegalaxy.org.au and usegalaxy.org[62]. The datasets selected to run BUSCO were the closest to the lineage of the species under study, eukaryota_odb10 and euglenozoa_odb10 datasets. The web resource SMART (Simple Modular Architecture Research Tool)[64] was used to corroborate the domain architecture of the putative MIPs. All sequences used for the phylogenetic analysis and the information about their accession and type of data are listed in Supplementary Data 2–4. Multiple sequence alignment (MSA) was performed with retrieved sequences using MAFFT, V7 (E-INS-i strategy, leaving gappy regions, Blosum62 as scoring matrix and MAFFT homologous option activated). Prokaryotic MIPs were included as they appeared to have a high amino acid sequence similarity (30%) to kinetoplastid MIPs (Supplementary Data 5) and appeared in BLAST searches when the Kinetoplastea class was excluded. Sequences were then trimmed using TrimAL (-g 0.8 -cons 50) to conserve only the more confidently aligned regions.

Phylogenetic trees were built using IQ-TREE[65] 2.0-rc2 and the evolutionary relationships among sequences were inferred by using the maximum likelihood (ML) method. The best-fit model was found using ModelFinder[66]. Branch support was calculated with the ultrafast bootstrap test[67] (10,000 iterations) and the Shimodaira-Hasegawa-approximate likelihood ratio test (SH-aLRT)[68] (1,000 iterations). The best-fit model was LG+F+R8 for the Prokaryotic MIPs analysis, LG+F+R7 for the Preliminary tree of Discoba MIPs and, LG+F+R6 for the Discoba AQPX tree. The phylogenetic tree files in newick format are provided in Supplementary Data 10–12. Trees were edited using the Interactive Tree of Life tool[69]. A visually revised alignment based on the resultant tree topology was constructed by manually correcting alignment errors and the phylogenetic tree analysis was performed again.

**Synteny analysis.** Synteny analysis was conducted by using BLAST+ (version 2.10.1+,[70]), SimpleSynteny software[71] and by exploring the TriTrypDB[72] genome browser. First, tBLAST was performed using the MIP and surrounding proteins found 10 Kb upstream and 10 Kb downstream. *T. cruzi* was used as a reference for protein sequences of AQPX alpha-gamma. *T. brucei* as a reference for GLP sequences. The genomes used as subjects in tBLAST search were the ones from *T. cruzi* (TcruziCLBrenerNon-Esmeraldo-like), *T. brucei* (TbruceiTREU927), *T. theileri* (TtheileriEdinburgh), *T. grayi* (TgrayiANR4), *T. congolense* (Tcongo-lenseIL3000_2019), *B. ayalai* (BayalaiB08-376), *L. major* (LmajorLV39c5), *P. confusum* (PconfusumCUL13), and *B. saltans* (BsaltansLakeKonstanz). The assembly status and metrics of these genomes were calculated using Quast v5.0.2[73] and are reported in the Supplementary Table S1. To calculate the coverage of the regions used for synteny analysis, the raw reads used for the assemblies (Supplementary Data 8) were mapped to the corresponding assembled genome using Bowtie2 with default parameters[74], and then the coverage analysis was performed using SAMtools[75]. These analyses were performed in the Galaxy public servers at usegalaxy.org.au and usegalaxy.org[62]. For *T. brucei* we recovered the MIPs region coverage from TriTrypDB genome browser (Jbrowser). For *T. congolense* we could not find the SRAs used for the assembly in any public database. So, we used reads of another WGS project of the same strain to estimate the coverage. Synteny was checked by manual inspection of the tBLAST result table. Genomic regions showing syntenic genes were selected, including 1 Kb before and after the first and the last gene in synteny, respectively. These regions were used as input for SimpleSynteny software. SimpleSynteny uses mainly two cutoff parameters to find syntenic genes, the e-value, and the query coverage, set to 0.01 and 10%, respectively.

**MIP residue assessment.** MSA was performed as described above, in this case, to identify typical MIP residues in specific alignment positions. We used Bioedit 7.2.5[76] to visualize and extract specific positions from the MSA. We specifically gather the information regarding: (i) Froger Positions, from 2 to 5, P1 was left out given that it remained a conflictive position in the MSA; (ii) both canonical NPA motifs; and (iii) selectivity filter residues located the second and fifth transmembrane domains (TMM2 and TMM5 respectively) along with two residues in loop E (LE1 and LE2). Already characterized MIPs (*Escherichia coli* GlpF and AqpZ, *Tb*AQP1, 2 and 3) were used to check the alignment and the identity of the defined positions. Residue assessment was shown related to phylogeny to confirm these critical positions within the evolutionary history of the analyzed MIPs.

**Reporting summary.** Further information on research design is available in the Nature Research Reporting Summary linked to this article.

## Data availability
The datasets generated during and/or analyzed during the current study are available from the corresponding author on reasonable request.

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

## Acknowledgements

The authors acknowledge Dr. Julius Lukeš, who kindly provided PhM-4, PhF-6, *Hemistasia phaeocysticola*, *Rhynchopus rumis* and *Sulcionema specki* transcriptome assemblies, Dr. Vyacheslav Yurchenko, who kindly provided *Blechomonas ayalai* SRAs, and Dr. Juan Pedro Liron for providing the computing power to build and visualize the SSN. This work was supported by the Agencia Nacional de Promoción Científica y Tecnológica (PICT 2017-0244 granted to K.A.).

## Author contributions

Conceived and designed the experiments: F.C.T., K.A. and A.R.F. Performed the experiments: F.C.T., A.R.F. and R.L. Analyzed the data: F.C.T., R.L., K.A. and A.R.F. Wrote the paper: F.C.T., K.A. and A.R.F. Jointly supervised this study: K.A. and A.R.F. All authors revised the manuscript.

## Competing interests

The authors declare no competing interests.
