## [Peer Review File · Communications Biology]

Reviewers' Comments:

Reviewer #1:

Remarks to the Author:

Interesting, well presented and for the most part well done analysis. I have few comments.

Unclear to start here with MIPs which seems to be interchangeable with AQP - please clarify. Also MIPs is used in the introduction, whereas AQP has been used in pretty much all other literature. Also, first para would be better suited after the second and third where the diversity of these proteins is discussed.

Exclusive use of SSN means that the AQPX is not, *sensu stricto*, a new family but a cluster. I have yet to fully understand SSN in terms of reliability and rigour compared to phylogenetic methods, where considerable effort underpins statistical assessment. With the SSNs in F1, many sequences are apparently unconnected to the large cluster 1, which makes no real biological sense if these are all derived from a single ancestral gene (which they are). Overall, not that clear what F1 and F2 provide.

F3 - Unreliable statistical analysis for so diverged a dataset and also no backbone support at the critical nodes defining AQPX and others. The massive overrepresentation of AQPX may also distort the reconstruction.

The rest of the MS is very good and well presented. I would have preferred two different phylogenetic methods being used as is common practise and more that 10000 replicate runs, but I doubt this would affect the conclusions.

There is a paper in press the also discusses this topic, albeit with less phylogenetic rigour.

Quintana, J.F., and Field, M.C., (2021) 'Aquaporins and pentamidine resistance in African trypanosomes.' *Parasitology* (in the press)

Reviewer #2:

Remarks to the Author:

Tesan et al. show an elegant phylogenetic analysis of aquaporins, which are proteins relevant to the physiology and drug susceptibility in trypanosomatid parasites. They performed a sequence similarity network analysis and found that trypanosome aquaporins (major intrinsic proteins, MIPs) are primarily placed in a previously undescribed family named AQPX, which share some similarity to prokaryotic MIPs. Although trypanosomatids MIPs are distributed between GLP and AQPX, synteny analysis indicates that *T. brucei* lost AQPXs, whereas *T. cruzi* retained AQPX and lost GLPs. In contrast, *Leishmania* spp. appears to have both AQPXs and GLPs. The authors argue that the AQPX family expanded before the origin of parasitism. The work begins with relevant analysis of aquaporins to a broad biology community interested in channels and phylogenetics. It became narrow as it deepened into the biology of kinetoplastid parasites. Nevertheless, the work might be of interest to the aquaporin and phylogenetic community beyond trypanosomes.

A few comments and concerns are indicated below:

1) The phylogenetic analysis depends on the available genome sequences. It is unclear from the text the extent of genome sequences available for the organisms studied. For the initial analysis of sequence similarity network, many MIP sequences are available (prokaryote and eukaryote), which might not be a problem. For *T. brucei*, *T. cruzi*, and *Leishmania* spp. genomes are also well described. However, for other organisms and downstream analysis, the authors relied on RNA-seq data. Suppose these sequences are coming from Illumina, which are often short reads with variable mapQs. How sure are the authors that they represent a reliable sequence dataset for the genes analyzed? The methods do not indicate metrics that inform the reader of the quality of the dataset used. This should be included in the methods and discussed. Moreover, what is the

coverage obtained in these RNA-seq datasets? If the sequences are ambiguous or of poor quality, this whole analysis falls apart. Hence, this information is essential for the conclusions and the reader's evaluation of the work.

2) For the synteny analysis, what are the genome coverage available for the regions studied? This is especially important for organisms which genomes are not so well studied, such as *P. confusum*. Is this organism's genome available? What is the source (qualitywise) of the sequences?

3) In the Abstract, Introduction and (lightly) in Results/Discussion, the authors state that AQPX originated before parasitism, an exciting finding. However, the statement is unclear throughout the manuscript! Do the authors mean "before parasitism" as stated? Or is this meant before parasitism in the kinetoplastida? What is the evidence for AQPX to originate before parasitism?

4) In line 109, the authors state GLPs are present only in trypanosomatids. This seems incorrect.

5) Line 123, "After clustering to 85% sequence identity and filtering by length..." is this amino acid identity? Indicate it in the text.

6) Many abbreviations are not described in the text, making the text difficult to understand to a less knowledgeable reader. For example, in the Abstract, AQPX was not defined. There are over ten different abbreviations that were not defined in the Introduction, including TSAR, PIP, GIP, TIP, NIP, etc. Please, define them.

7) Introduction, line 94: the word "sanitary" refers to diseases caused by hygiene conditions, which is not the case of trypanosomatids of medical importance; they cause vector-borne diseases.

8) It is interesting that *T. brucei* lost AQPXs and retained GLPs, whereas *T. cruzi* lost GLPs and kept AQPXs. Are there other similar examples of unusual gene family loss/retention in these parasites? It would enrich the manuscript if the authors would speculate potential reasons or discuss similar examples. The discussion seems overall quite focused on trypanosomes and lacks some broad discussions of this type of events.

Reviewer #3:

Remarks to the Author:

The ms by Tesan et al. describes a theoretical analysis of a large number of aquaporin (AQP) protein sequences of the kinetoplastids using sequence similarity networks (SSN) and phylogenetic trees. As the main result the authors combine the previously termed AQP alpha/beta/gamma/delta types of sequences to a cluster, which they term AQPX. Aquaporins are channels for water and small uncharged solutes, mainly glycerol and urea. However, other small neutral molecules can pass, such as ammonia, hydrogen peroxide, or even protonated short-chain monocarboxylic acids. The variety is large due to limited selectivity capacities of the AQPs that is based alone on charge exclusion and size selection in the selectivity filter region.

From an evolutionary point of view the provided analysis may be of interest, namely the notion that African and American trypanosomes express a shifted profile of AQPs. In terms of relevance for drug treatment as suggested by the authors, this probably does not hold true, though. In this regard and concerning selectivity mechanisms, the ms contains some errors, and outdated or one-sided views, which should be corrected.

1. I37, and others: The wording "transport" must not be used in connection with channel functionality. Please correct throughout the ms.

2. I41, "previously undescribed family": the authors give a literature reference to this type of AQP isoforms, hence it has been described.

3. I68: The functional role of diverse NPA motifs is overrated throughout the ms, probably because

statements are based on older references. The main proton filter resides in the "selectivity filter" region, see Wu et al. EMBO J. 2008 and Kosinska Ericsson et al. Science 2013.

4. I387: variations in the NPA motifs at the second and third position, e.g. NSA/NPS, have never been shown to have consequences on the selectivity. The relevant position is the "N".

5. I393: the matter whether pentamidine can actually pass TbAQP2 is not clear. An alternative model explains TbAQP2-dependent uptake of pentamidine by high-affinity binding to a uniquely exposed aspartate in the selectivity filter "D" (see point 6) and endocytosis of the complex, see Song et al. PLoS Pathog. 2016.

6. I.427: The specialty of TbAQP2 is the "L" in the "R" position of the selectivity filter IN COMBINATION with keeping the "D" right next to the "R". The named other AQPs lacking a positively charged residues in the selectivity filter also have a neutral residue in the following "D" position. This "D" or absence thereof should be integrated into assumptions on structure-function relationships.

Communications Biology - Response to reviewers

Article Title: **Aquaporins belonging to the newly described cluster AQPX and aquaglyceroporins are asymmetrically distributed in trypanosomes.**

Authors: Fiorella Carla Tesan, Juan Ramiro Lorenzo Lopez, Karina Alleva, Ana Romina Fox.

❖ **A point-by-point response to the referee's comments is shown below. Manuscript changes are highlighted in red color.**

N°	Referee #1's comments	Author's reply
	Interesting, well presented and for the most part well done analysis. I have few comments.	
1	i- Unclear to start here with MIPs which seems to be interchangeable with AQP - please clarify. Also MIPs is used in the introduction, whereas AQP has been used in pretty much all other literature. ii- Also, first para would be better suited after the second and third where the diversity of these proteins is discussed	i- In this manuscript we did not use the terms MIP and AQP as interchangeable terms, we tried to be clear on this (Introduction, lines 79-81). But we agree with the reviewer that sometimes the term AQP has been used to refer to the MIP superfamily by the academic community. Also, mammals MIPs are named AQP1-13 without distinguishing AQPs from GLPs. Notwithstanding, we will comment here on the problematic nature of this nomenclature and the reasoning for our stand on the topic. In the early '90s, the transmembrane channels of water and other solutes were described as homologous proteins (by sequence similarities) and were grouped and named as part of the MIP family, integrated then by 20 members (Pao et al., 1991; Reizer et al., 1993; Chrispeels and Agre 1994). At that time, the aquaporins were a functionally defined group of water-transporting channels (Chrispeels and Agre, 1994), so the term AQP was reserved for that specific transport function. By the year 2001, the homology assignment was based on phylogenetic inferential methods using more than 150 members of the MIP family (Zardoya et al., 2001). In this work, both the terms MIP and AQP were used to name the superfamily and, simultaneously, the term AQP was used to name a specific phylogenetic cluster of water channels (Zardoya et al., 2001). More than a decade later, some studies continue using the terms AQP and MIP as synonymous though they acknowledge the AQP as a specific cluster (Perez Di Giorgio et al., 2014; Finn et al., 2014), whereas others mention that “Membrane intrinsic proteins (MIPs) are also termed aquaporins” but carry on reserving the term AQP for the specific clade an MIP for the whole superfamily (Abascal et al., 2014). Recently, other classifications (wAQP, for water-selective aquaporins; gAQP for bacterial glycerol channel-like aquaglyceroporin; and sAQP, for intracellular super aquaporins) were proposed (Ishibashi et al., 2020), but their use in the community is not yet expanded. Additionally, all protein databases acknowledge signatures for these proteins and group all of them among the MIP superfamily (Pfam, Interpro, CDD, OPM, PROSITE, etc). Altogether, we believe that there is no ambiguous interpretation of the term MIP but that is not the case for the term AQP. Therefore, in our manuscript, we left the term AQP for the specific group though it is recognized as polyphyletic (Finn et al., 2014), and the term MIP to refer to the whole superfamily. ii- We appreciate the reviewer's suggestion and so we worked on the proposal of

		reordering the paragraphs of the introduction. We assayed several versions but unfortunately, we were not convinced by the results of any of them. Instead, we prefer to present the organism targets of our research and to explain why they are of interest in the first paragraph. We believe this brief initial mention makes the understanding of the information provided in the rest of the introduction, easier. Then, over the following paragraphs, we focus on the actual knowledge on MIPs diversity including a detailed survey on MIP isoforms of the Trypanosomatida order (Kinetoplastids, Discoba). So, we would like to preserve the original structure of the Introduction, as it does not involve a serious comprehension problem and suits our intended meaning better.
2	i- Exclusive use of SSN means that the AQPX is not, sensu stricto, a new family but a cluster. ii- I have yet to fully understand SSN in terms of reliability and rigour compared to phylogenetic methods, where considerable effort underpins statistical assessment. With the SSNs in F1, many sequences are apparently unconnected to the large cluster 1, which makes no real biological sense if these are all derived from a single ancestral gene (which they are). Overall, not that clear what F1 and F2 provide.	i- This is true if we do not consider the phylogenetic studies and, for this reason, the cluster found in the network was named with a subindex (AQPX_{SSN}). MIPs most likely share a single ancestor among prokaryotes, but the superfamily already diversified in prokaryotes. Different prokaryotic MIPs share a common ancestor with different eukaryotic MIPs and, for that reason, it is considered a polyphyletic superfamily (Finn et al., 2014; Pommerrenig et al., 2020). We performed a phylogenetic analysis of prokaryotic MIPs to analyze this issue (Results and discussion, lines 169-177) before proposing AQPX as a new family already present in prokaryotes. ii- The assignment of family or subfamily members by phylogeny approaches relies on a single and accurate multiple sequence analysis (MSA). Non-redundant sequences of protein databases are rising, and the computational complexity of MSA-based phylogenies is exponentially increased with the number of sequences. Hence, phylogenetic analysis of large protein families becomes unviable (Viborg et al., 2019). In those scenarios, SSN can assist phylogenetic analysis, as shown in this work. Since SSN is based on all-versus-all pairwise local sequence alignments, and it does not suppose any evolutionary model, it has fewer computational requirements. Then, SSN can highlight unexplored sequence space (Copp et al., 2018) as was performed here. Some trypanosomatid MIP sequences alone (those we then named AQPXs) look different than the rest of the characterized MIPs and we were not comfortable arbitrarily selecting some of the characterized MIPs as roots of the phylogenetic trees because their amino acid sequence identity was very low no matter the phylogenetic distance of the organism. So, we decided to place them in the sequence-function space of the whole protein superfamily otherwise impossible by manual or one by one analysis. And there lies the power of this tool. In this way, the results from the SSN allowed us to understand why it was not easy to find a root for trypanosomatid AQPs. We found these AQPs to cluster with uncharacterized prokaryotic sequences, and some protozoan sequences, in the cluster AQPX_{SSN}. This finding (F1 and F2) opened the way to the study of the AQPXs as a family. Regarding the referee's concern about the unconnected nodes, we set the cutoff values for SSN branches in an alignment score of 35 (corresponding to 35-40% pairwise sequence similarity). When the alignment score was lower the nodes were unconnected. This threshold value is selected arbitrarily to separate clusters. If you use a very low value, you will get one very big cluster. In the other extreme, if you use a very high threshold value, the “isofunctional” families will be fragmented. In the case of the MIP superfamily, we have a lot of information regarding phylogenetic clusters, we can

		observe them in the SSN and use them to assist us about which threshold value to choose. But the selected threshold defines nothing about sharing a common ancestor or not. That is why there are clusters not connected to the main one (cluster 1). Higher cutoff values would break cluster 1 or even subclusters, but that has no phylogenetic information, as commented by the referee.
3	i- F3 - Unreliable statistical analysis for so diverged a dataset and also no backbone support at the critical nodes defining AQPX and others. ii- The massive overrepresentation of AQPX may also distort the reconstruction. iii- The rest of the MS is very good and well presented. I would have preferred two different phylogenetic methods being used as is common practise and more that 10000 replicate runs, but I doubt this would affect the conclusions.	i- This first phylogenetic analysis was exploratory, and we knew that we were including MIPs that diversified before the origin of the eukaryotic life (at least GLPs from AQPs). But we wanted to show a detailed picture of Discoba MIPs diversity. For that reason, we took the idea from Finn et al. (2014) of calling this first tree “preliminary” as we are using a “divergent dataset” consequence of the polyphyletic nature of the superfamily. Through this tree, we expose several features of Discoba MIPs diversity that highlight the particularity of Kinetoplastid MIPs as it is discussed in the manuscript (Results and discussion, lines 229-272). Importantly, this analysis, together with the SSN, is useful to discard the previous idea that trypanosomatid AQPs are TIP-likes. Regarding the statistical support, the tree was rooted in the long and fully supported branch that separated GLPs from the other MIPs, and the AQPX cluster is also well supported by both SH-aLRT and UF-BS. ii- As a supplementary file we showed how AqpXs integrate an uncharacterized grade of Prokaryotic MIPs. In that phylogenetic inference, there is not a massive overrepresentation of AqpXs and the different grades of MIPs are observed and well supported. For the preliminary tree of Discoba MIPs we picked some bacterial isoforms of each grade and added them to our analysis to help us to identify clusters. Bacterial Glps can be observed as an outgroup of the main GLP group, Bacterial AqpN grouped with Andalucia godoyi’s MIPs that were previously described as NIP-like and, Bacterial AqpX grouped with Discoba AQPXs. Then, it seems to us that there is no distortion of the tree consequence of the abundance of AQPXs. Even though, to discard the hypothesis of an overrepresentation of AQPXs we performed again the analysis discarding all the bodonids and trypanosomatids but B. saltans, T. brucei and T. cruzi. The 65 Discoba MIP sequences left lead to a phylogenetic tree having the same 1 to 9 group distribution as the manuscript’s preliminary tree did (Revision Figure 1, added at the end of this referee’s response). This new phylogenetic inference included a comparable number of isoforms of the MIP families present in trypanosomatids: 19 out of the total 65 sequences were AQPX isoforms and 13 were GLPs. Each of the referenced groups is well supported and therefore we have no reason to believe there is a distorted reconstruction. iii- The use of one phylogenetic method can be found in other publications regarding trypanosomatid proteins (De Souza et al., 2018; Jackson et al., 2016) and even MIP global phylogeny (Zardoya 2005; Abascal et al., 2014). Also, specifically for MIP evolution we can observe that the global tree architecture is conserved when the tree is constructed using ML -Fig. 4b (Abascal et al., 2014)- or using Bayesian inference -Fig. 2, (Finn et al., 2014)- (those two trees include Metazoan species, the MIP sequences are not the same and the trees are constructed by different authors and still they found the same connections among MIP clusters). Regarding our trees, of the different available platforms to perform ML analysis we choose IQ-tree that compares favorably to RAxML and PhyML in terms of likelihood

		(Nguyen et al., 2015). Also, through the option ModelFinder, we selected the model of sequence evolution that best fitted our data (54 protein models were tested). Regarding replicate runs to estimate BS support, the IQ-Tree developers of the UF-BS method advise using 1000 replicate runs as a minimum (http://www.iqtree.org/doc/Tutorial), being this number a conservative upper bound to achieve high accuracy (Minh et al., 2013). And you can find ML phylogenetic analysis using UF-BS with 1000 replicates (Imhoff et al., 2018; Butenko et al., 2020). According to all this information, we decided to run 10 times more replicates than recommended and commonly used (10,000). Finally, the general structure of the AQPXs tree is congruent with the tree of life with well-supported nodes. Then, we agree with the referee that using another phylogenetic method would difficulty affect our conclusions.
4	There is a paper in press the also discusses this topic, albeit with less phylogenetic rigour. Quintana, J.F., and Field, M.C., (2021)'Aquaporins and pentamidine resistance in African trypanosomes.' Parasitology (in the press)	We thank the referee for facilitating this article. This review recovers the already published data on trypanosome MIPs. Indeed, the authors name the AQPs of T. cruzi and Leishmania spp. as TIP-like AQPs because, currently, that is the only proposed phylogenetic relation (Introduction, lines 109-110). Our analysis of the MIP superfamily (SSN and preliminary phylogenetic analysis) does not support that first proposal that was based on phylogenetic analysis in which one GLP (Escherichia coli), one plant TIP (Arabidopsis thaliana), and one metazoan AQP (Homo sapiens) have been used as references. Moreover, AQPXs have a rare pattern of SF that does not resemble the SF of TIPs (except for the V in LE2 position of some TIPs -TIP1-like MIPs-) as we discussed in the manuscript (Results and discussion, lines 477-481). We included a reference to this recent publication (Introduction, line 110).

Revision Figure 1. Preliminary tree of the Discoba supergroup MIPs. Phylogenetic tree was built with 69 Discoba MIP sequences and tree topology was reconstructed by maximum likelihood. Branch support was assessed by ultrafast bootstrap (UF) and the SH-aLRT approximations with 1,000 replicates. Support above 80 for SH-aLRT and/or above 90 for UF is shown on each branch (SH-aLRT/UF). Each MIP clade is collapsed (triangles) and referenced in correspondance to original Preliminary tree Groups 1 to 9 (Figure 3 of the ms). MIP sequence number of each clade is shown inside representative triangles.

N°	Referee #2's comments	Author's reply
		Tesan et al. show an elegant phylogenetic analysis of aquaporins, which are proteins relevant to the physiology and drug susceptibility in trypanosomatid parasites. They performed a sequence similarity network analysis and found that trypanosome aquaporins (major intrinsic proteins, MIPs) are primarily placed in a previously undescribed family named AQPX, which share some similarity to prokaryotic MIPs. Although trypanosomatids MIPs are distributed between GLP and AQPX, synteny analysis indicates that T. brucei lost AQPXs, whereas T. cruzi retained AQPX and lost GLPs. In contrast, Leishmania spp. appears to have both AQPXs and GLPs. The authors argue that the AQPX family expanded before the origin of parasitism. The work begins with relevant analysis of aquaporins to a broad biology community interested in channels and phylogenetics. It became narrow as it deepened into the biology of kinetoplastid parasites. Nevertheless, the work might be of interest to the aquaporin and phylogenetic community beyond trypanosomes. A few comments and concerns are indicated below:
1	i- The phylogenetic analysis depends on the available genome sequences. It is unclear from the text the extent of genome sequences available for the organisms studied. ii- For the initial analysis of sequence similarity network, many MIP sequences are available (prokaryote and eukaryote), which might not be a problem. For T. brucei, T. cruzi, and Leishmania spp. genomes are also well	i- As exposed in Material and Methods (lines 541-547), the first searches were performed in public repositories (NCBI, TritypDB and iMicrobe), and the databases explored in those searches were not fully detailed in our manuscript (especially those from the NCBI). To better expose sequence availability and clarify what the referee remarked we split the original Supplementary Data 2 into two Supplementary Data files. The new Supplementary Data 2 is a complete report of the sequence databases analyzed and includes a metric to evaluate the genomes/transcriptomes completeness (this topic will be discussed below). And the new Supplementary Data 4 holds MIPs details for each of the analyzed species (number of MIPs, identifiers, and positions in the preliminary phylogenetic tree). This split simplified the reading of the information. We added the references to these Supplementary Data in pertinent manuscript positions (Results and discussion, lines 200-201). We performed an intensive search of MIPs in publicly available databases and stumbled upon heterogeneous genome sequence availability (detailed in Supplementary Data 2). (Results and discussion, lines 217-218) The complete list of MIPs here analyzed is reported in Supplementary Data 4. Modification of Supplementary Data titles: Supplementary Data 2. Genomes and transcriptomes of Discoba organisms analyzed in this study. Supplementary Data 4. MIP identifiers and position in the preliminary phylogenetic tree. ii- Most of the transcriptomes here analyzed have already been used in other evolutionary studies from where we got evidence of their good quality. The transcriptomes of Trypanoplasma borrelli, Neobodo designis, Azumiobodo hoyamushi, PhM-4, PhF-6, Hemistasia phaeocysticola, Rhynchopus humris, Sulcionema specki, Euglena gracilis, and Rhabdomonas costata were successfully used to carry out a comparative analysis of euglenozoans metabolic enzymes and molecular features (DNA pre-replication complex, kinetochore machinery) (Butenko et al., 2020). And the same transcriptome database of Azumiobodo hoyamushi was used in a phylogenomic study of Kinetoplastea in which a 43-gene alignment was performed and all A. hoyamushi

described. However, for other organisms and downstream analysis, the authors relied on RNA-seq data. Suppose these sequences are coming from Illumina, which are often short reads with variable mapQs. How sure are the authors that they represent a reliable sequence dataset for the genes analyzed? The methods do not indicate metrics that inform the reader of the quality of the dataset used. This should be included in the methods and discussed. Moreover, what is the coverage obtained in these RNA-seq datasets? If the sequences are ambiguous or of poor quality, this whole analysis falls apart. Hence, this information is essential for the conclusions and the reader's evaluation of the work.	orthologs were present (Yazaki et al., 2017). Still, some transcriptomes may not represent reliable databases. For example, those of Diplonema ambulator and Rhynchopus euleeides, even if they were used to analyze mitochondrial transcripts, we cannot assume that they represent a complete database (Valach et al., 2017). Additionally, the transcriptomes of the Percolomonas cosmopolitus strains are not yet associated with any publication. Therefore, we agree with the referee that it is better to show metrics that inform the quality of the dataset. We chose the tool BUSCO (https://busco.ezlab.org/) (Simão et al., 2015) that provides a quantitative estimation of the completeness in terms of the expected gene content, of a genome or transcriptome assembly. The program provides a % of complete and fragmented universal orthologs present in your dataset (this will be discussed and added to the manuscript as indicated below). On the other hand, there are four organism databases not analyzed with BUSCO: Parabodo caudatus, Procryptobia sorokini, Reclinomonas americana, and Seculamonas ecuadoriensis. P. caudatus and P. sorokini MIPs were retrieved for transcriptomes in where these organisms were prey (Results and discussion, lines 205-207) and they were assigned to these preys comparing the in-home assemblies with the prey-clean published assemblies. Also, these MIPs were blasted against the NCBI databases and both P. caudatus and P. sorokini MIPs' best BLAST hits are from Metakinetoplastina organisms. In the case of P. caudatus MIPs, some of them were identical to predicted MIPs of C. angusta (TSAR) coming from transcriptomes in which P. caudatus was a prey. All this information is present in Material and Methods (lines 548-556) and Supplementary Data 3. Also, we are confident that we are not wrongly assigning sequences because in both cases sequence positions in the phylogenetic tree were congruent with the tree of life. For R. americana, and S. ecuadoriensis, MIPs were retrieved from EST submitted to the NCBI database and the EST libraries are not available. So, for these two organisms, we replaced the word transcriptome for EST in Supplementary Data 2 to be more precise regarding the origin of the sequences. To search for universal orthologs there are different lineage datasets in BUSCO. The rule is to use the one that better represents your species of interest. We tested two of them: eukaryota_odb10 and euglenozoa_odb10. The second one theoretically may provide a good dataset for the analyzed transcriptome/genomes. But from our results, we can see that the euglenozoans dataset is good only for trypanosomatids and Bodo saltans without providing a good representation of the complete phylum. For the rest of the kinetoplastids, it is more or less the same to use one or the other dataset whereas, for diplomemids or euglenids the eukaryota_odb10 provides higher scores. Looking inside the database we can see that euglenozoa_odb10 was constructed only with trypanosomatids, Bodo saltans, and Perkinsela, and that may be the reason for the bias in the analyses. Thus, we reasoned that a good compromise would be to integrate the data of the two BUSCO analyses. Using the eukaryota_odb10, the BUSCO scores for trypanosomatids, Neobodonida and Prokinetoplastina organisms are in the same range of values (40-60%, excluding Perkinsela that it is known to have a really small genome). Diplonemids and Euglenids have scores over 60%, being reasonable values for transcriptomes. Naegleria is the best known and most studied genus within Heterolobosea. There are genomes and transcriptomes with high scores for this genus
---	--

but none AQPX was found in there. Instead, we found AQPXs in another genus, *Percolomonas*. The BUSCO scores for the transcriptomes of two strains of *P. cosmopolitus* were not high and neither homogeneous, but we found one AQPX in each assembly. We cannot say anything about what is absent in these transcriptomes, but we do not have reasons to doubt the sequences found in our analysis.

Finally, we agree with the referee that the databases need to be reliable. But we think that with the transcriptomes used here the biggest interpretation problems may not arise from poor coverage but contamination with prey organisms used in the cultures, then we also centered our attention on this, and we add a paragraph about *Percolomonas* transcriptomes (Material and Methods, lines 559-566). The second point is that we cannot expect to get all the possible MIPs from a transcriptome. Even when a transcriptome may be complete, for sure there is more than one transcriptome for each genome. MIP expression may vary among environmental conditions for example. And, being aware of this fact, most of our conclusions are based in the presence of MIPs, not in their absence.

The manuscript was modified as follows:

Results and discussion (addition in lines 273-282: new, independent paragraph):

Finally, to analyze the reliability of the heterogeneous sources of *Discoba* MIPs we analyzed the completeness of the genome and transcriptome assemblies using the tool Benchmarking Universal Single-Copy Orthologs (BUSCO). Most of them showed good levels of completeness (Supplementary Data 2, analyzed in Supplementary Results and Discussion). Additionally, most of the transcriptomes here analyzed were already used to successfully carry out a comparative analysis of euglenozoans metabolic enzymes and molecular features (DNA pre-replication complex, kinetochore machinery) (Butenko et al., 2020). Altogether, this indicates that a reliable set of assemblies were used in our MIPs searches. Still, it is worth mentioning that a different picture might be reconstructed once more *Discoba* organisms have their genomes sequenced and can be included in the study.

Supplementary Results and discussion (addition of new paragraph, under the subtitle *Analysis of Discoba genomes and transcriptomes completeness*):

BUSCO provides a quantitative estimation of the completeness of the assemblies in terms of expected gene content using as reference an OrthoDB set of “universal” orthologs (Simão et al., 2015). BUSCO provides different lineage datasets of orthologs and we tested two of them: eukaryota_odb10 and euglenozoa_odb10. The euglenozoa_odb10 provides high scores (77-100%) only for trypanosomatids and *Bodo saltans*. For the rest of the kinetoplastids, it was more or less the same to use one or the other dataset, whereas for diplomemids and euglenids, the eukaryota_odb10 provides higher scores (Supplementary Data 2). The reason for the bias in the analyses is that the euglenozoa_odb10 was built only with trypanosomatids, *Bodo saltans*, and *Perkinsela* reference genes. Thus, we reasoned that a good compromise was to consider the BUSCO results obtained using the eukaryota_odb and to consider that the trypanosomatids that are completely sequenced got scores ranging from 40 to 56% using this dataset. Therefore, the transcriptomes of *Neobodonida* and *Prokinetoplastina* that got scores

		ranging from 40 to 64% -excluding Perkinsela that is known to have a small genome (Tanifuji et al., 2017)- can be considered representative for these species. Diplonemids and Euglenids scores were over 60% of completeness, being reasonable values for transcriptomes. The genus Naegleria, the best known and most studied genus within Heterolobosea, got high scores (over 75% complete). Finally, the Percolomonas cosmopolitus strains got scores of 35 and 54%. Then, from these assemblies, any suggestion of absence of genes should be taken with caution. Material and Methods (addition in lines 559-570) Percolomonas cosmopolitus cultures were fed with Enterobacter aerogenes. Thus, the presence of bacterial contaminating transcripts was tested for the strain WS. Megablast of Percolomonas cosmopolitus Strain WS assembly against BLAST nucleotide database (nt17-Apr-2014) showed that 524 of 11,058 query sequences had a match with sequences found in the database (cut off e-value 10^{-3}). When selecting only the first match for the 524 query sequences (hit with lowest e-value), 26 query sequences were matching with bacterial sequences. Suggesting very low contamination with bacterial RNA (26/11,058) and none of the matches correspond to the MIPs found in the transcriptome. The quality and completeness of the proteomes, transcriptomes, and genomes used in this study were assessed by using BUSCO tool suite v5.0.0 (Simão et al., 2015) in the Galaxy public servers at usegalaxy.org.au and usegalaxy.org (Afgan et al., 2018). The datasets selected to run BUSCO were the closest to the lineage of the species under study, eukaryota_odb10 and euglenozoa_odb10 datasets.
2	For the synteny analysis, what are the genome coverage available for the regions studied? This is especially important for organisms which genomes are not so well studied, such as P. confusum. Is this organism's genome available? What is the source (quality wise) of the sequences?	The genome of Paratrypanosoma confusum is available at TritrypDB and it was sequenced by Skalický et al., 2017 (Skalický et al., 2017). The authors reported a complete genome assembly in which most core eukaryotic genes were present. Consistently, our BUSCO analysis provides scores similar to those of the other trypanosomatids (Supplementary Data 2). We added to the manuscript a Supplementary Table 1 to provide the readers access to the assembly status for the nine genomes analyzed in the synteny study. The table also provides metrics of the quality of the genomes. Four of the nine genomes are assembled at the chromosome level, two at the supercontig level (among these P. confusum) and three at the contig level. For T. brucei we recovered the MIPs region coverage from TriTrypDB genome browser (Jbrowser). For T. theileri, T. grayi, L. major, B. ayalai, P. confusum, T. congolense and B. saltans we aligned the reads used for the genomes sequencing to the published genomes and calculated the average coverage for each region. For T. congolense we could not find the SRAs used for the assembly in any public database. So, we used reads of another WGS project of the same strain to estimate the coverage. The genome assembly to the chromosome level of T. cruzi strain Brener non-Esmeraldo-like resulted from the integration of different WGS projects, so there is not one unique source of SRAs that we can use to calculate the coverage (Weatherly et al., 2009). Then, for T. cruzi the coverage was not calculated. We added to Supplementary Figures 3-8 the coverage calculated for each region. Additionally, we added the position of Ns (unknown sequences) in Supplementary Figures 3-8, even if this is not a metric, it is

useful information to analyze the quality of the assemblies on the specific regions.

We would like to mention some cases of unknown sequence fragments: 1- the AQPalpha region for the genome of *T. theileri*, and 2- AQPbeta-delta and AQP2-3 genomic regions of *P. confusum*.

1- In the synteny analysis of AQPalpha *T. cruzi* is the reference genome. The *T. theileri* AQPalpha region is present in a small contig of 5,000 bp together with two up-stream genes orthologs to TcAQPalpha up-stream neighbours. The genes syntenic with the down-stream genes of TcAQPalpha localize in a much bigger contig (1.6 million bp), and the physical position expected for *T. theileri* AQPalpha and the upstream neighbours is missing. Instead in this region there are Ns. In this case the most probable and parsimonious explanation is that *T. theileri* and *T. cruzi* AQPalpha regions are in synteny but the region is not well defined for *T. theileri*.

2- *P. confusum* average genome coverage and the coverage of the specific regions are the lowest among the analyzed genomes and this matches with a higher quantity of N regions. Two N regions are present in MIP expected regions of the genome of *P. confusum*: i-AQPbeta-delta genomic region and, ii- AQP2-3 genomic region. For the AQPbeta-delta genomic region the situation is similar to that of *T. theileri* AQPalpha region. The PconAQPdelta is present in a small non assembled contig while a region of Ns is present in a larger scaffold at the AQP expected site.

For the region in which we “expected” to find AQP2-3 of *P. confusum* the situation is different. This genomic region is not syntenic among trypanosomatid subfamilies (lines 378-379) and the region is not defined for *P. confusum* (plenty of Ns). Still, the genes up- and down-stream of the GLPs position seem to be in synteny among *T. brucei*, *P. confusum* and *B. saltans*.

We found no orthologs of *TbAQP2* or *TbAQP3* coded on the genome of *P. confusum*, but the undefined region can place the doubt of whether the GLPs are present on its genome. Therefore, we assembled 8 SRAs of RNA-seq experiments available for different stages of *P. confusum* at the NCBI and published in Skalický et al., 2017, we found one GLP in one of the SRA experiments (SRR6231228). But the sequence was nearly identical to a GLP of *Leptomonas pyrrhocoris* (99.1 % cDNA identity, 98% amino acid identity with LpyrH10_32_1200). Thus, we blasted the assemblies against the genomes of *P. confusum* and *L. pyrrhocoris* and clearly there is some technical issue of transcriptome contamination in that particular assembly where we found the GLP and in other one (Revision Table 1, at the bottom of the answers to this referee, in bold and blue the problematic SRAs. MegaBLAST cut-off at 99% sequence identity). Therefore, we discarded the two SRA experiments with high % of hits with *L. pyrrhocoris* (SRR6231227/8) in the analysis of *P. confusum* transcripts. Finally, we found no GLPs on *P. confusum* genomes or transcriptomes. To complete the analysis, we also assembled transcriptomes of *B. saltans* to look for GLPs and double check their absence in this organism. We found no GLPs coded on their transcriptomes.

We added part of this analysis to the manuscript as follows:

Results and discussion (additions)

		(in lines 331-337) We analyzed nine genomes, four of them are assembled at the chromosome level (T. cruzi, T. brucei, T. congolense and L. major), two at the supercontig level (P. confusum and B. saltans) and three at the contig level (T. grayi, T. theileri and B. ayalai) (Supplementary Table 1). Overall, the quality of the assemblies, even if not homogeneous, is undoubtedly good. The genome coverages for the studied regions are among 41x and 200x (Supplementary Data 8). The coverage and undefined regions (Ns) are available in Supplementary Figures 3-8. (in lines 385-389) Moreover, this large region of P. confusum is undefined and therefore we cannot exclude the presence of a GLP in there. Therefore, we assembled transcriptomes available for P. confusum (Supplementary Data 2) and searched for GLPs, finding none. To complete the analysis, we also searched for GLPs in B. saltans transcriptomes (Supplementary Data 2) and we found none either. Material and Methods (lines 600-609) The assembly status and metrics of these genomes were calculated using Quast v5.0.2 (Gurevich et al., 2013) and are reported in the Supplementary Table S1. To calculate the coverage of the regions used for synteny analysis, the raw reads used for the assemblies (Supplementary Data 8) were mapped to the corresponding assembled genome using Bowtie2 with default parameters (Langmead and Salzberg, 2012), and then the coverage analysis was performed using SAMtools (Danecek et al., 2021). These analyses were performed in the Galaxy public servers at usegalaxy.org.au and usegalaxy.org (Afgan et al., 2018). For T. brucei we recovered the MIPs region coverage from TriTrypDB genome browser (Jbrowser). For T. congolense we could not find the SRAs used for the assembly in any public database. So, we used reads of another WGS project of the same strain to estimate the coverage. Acknowledgments (lines 819-822) Before: The authors acknowledge Dr. Julius Lukeš, who kindly provided PhM-4, PhF-6, Hemistasia phaeocysticola, Rhynchopus rumis, and Sulcionema specki transcriptome assemblies and Dr. Juan Pedro Liron for providing the computing power to build and visualize the SSN. Now: The authors acknowledge Dr. Julius Lukeš, who kindly provided PhM-4, PhF-6, Hemistasia phaeocysticola, Rhynchopus rumis, and Sulcionema specki transcriptome assemblies, Dr. Vyacheslav Yurchenko, who kindly provided Blechnomonas ayalai SRAs, and Dr. Juan Pedro Liron for providing the computing power to build and visualize the SSN.
3	In the Abstract, Introduction and (lightly) in Results/Discussion, the authors state that AQPX originated before parasitism, an exciting finding. However, the	Throughout the manuscript, we intend to state that the AQPX family expanded before the origin of parasitism in the Metakinetoplastina common ancestor (Abstract, lines 45-46; Introduction, lines 120-121). We meant that AQPX expanded (not originated) before parasitism but referring specifically to the expansion that originates the α-δ clades present among the parasites of the Trypanosomatida order (Results and discussion, lines 309-311). Based on the finding that all metakinetoplastina species hold up to four AQPX in the α-δ clusters and

	statement is unclear throughout the manuscript! Do the authors mean "before parasitism" as stated? Or is this meant before parasitism in the kinetoplastida? What is the evidence for AQPX to originate before parasitism?	that prokinetoplastina species hold AQPX isoforms that appear only in a sister clade (or none, as Perkinsela sp.), we placed the expansion of this family in the metakinetoplastina ancestor, before Bodo saltans branched and therefore before the origin and diversification of extant parasitic trypanosomatid species. Having into account that parasitism evolved several times independently among kinetoplastids we agree that we need to be more specific to not induce wrong ideas. We apologize for the confusion and we carefully went through the manuscript to detect those phrases that were not clear and modified them as detailed below. Abstract (lines 44-47) Before: Our phylogenetic analyses reveal that trypanosomatid MIPs distribute exclusively between aquaglyceroporin (GLP) and AQPX, being the AQPX family expanded before the origin of parasitism, in the Metakinetoplastina common ancestor. Now: Our phylogenetic analyses reveal that trypanosomatid MIPs distribute exclusively between aquaglyceroporin (GLP) and aquaporin X (AQPX), being the AQPX family expanded in the Metakinetoplastina common ancestor before the origin of the parasitic order Trypanosomatida. Introduction (lines 116-122) Before: In this work, we show that two MIP families expanded among trypanosomatids: GLP and a previously undescribed MIP family, named here AQPX. While GLPs are present only in trypanosomatids, the AQPX family expanded before the origin of parasitism, and extant trypanosomes hold up to four AQPX paralogs. Now: In this work, we show that two MIP families expanded among trypanosomatids: GLP and a previously undescribed MIP family, named here AQPX. GLPs were not found in other kinetoplastid orders, whereas AQPXs were found in early-branching kinetoplastids. The AQPX family expanded in the Metakinetoplastina common ancestor before the origin of the parasitic order Trypanosomatida and extant trypanosomes hold up to four AQPX paralogs.
4	In line 109, the authors state GLPs are present only in trypanosomatids. This seems incorrect.	This sentence was corrected (Introduction, lines 118-119) with the referee comment N° 3. Before:...GLPs are present only in trypanosomatids, Now:...GLPs were not found in other kinetoplastid orders...
5	Line 123, "After clustering to 85% sequence identity and filtering by length..." is this amino acid identity? Indicate it in the text.	Yes, it is the amino acid identity percentage. Results and discussion (line 133) Before: After clustering to 85% sequence identity and filtering by length, 16,170 representative accessions composed the network's nodes. Now: After clustering to 85% amino acid sequence identity and filtering by length, 16,170 representative accessions composed the network's nodes.
6	Many abbreviations are not described in the text,	We have now defined all the abbreviations missing in the text as follows:

	making the text difficult to understand to a less knowledgeable reader. For example, in the Abstract, AQPX was not defined. There are over ten different abbreviations that were not defined in the Introduction, including TSAR, PIP, GIP, TIP, NIP, etc. Please, define them.	Abstract (lines 43-44) Here we place trypanosomatid channels in the sequence-function space of the large MIP superfamily and expose their presence in a previously undefined family, here named aquaporin X (AQPX). Introduction (lines 85-91) In Eukarya, there are up to seven recognized families of land plant MIPs: plasma membrane intrinsic protein (PIP), tonoplast intrinsic protein (TIP), Nodulin 26-like intrinsic protein (NIP), small basic intrinsic protein (SIP), X-intrinsic protein (XIP), hybrid intrinsic protein (HIP), and GlpF-like intrinsic protein (GIP), while green algae have PIPs and GIPs but also other five subfamilies (named MIP A–E) not found in land plants. (lines 100-102) In contrast, little is reported regarding other supergroups, such as Discoba, TSAR (Telonemia, Stramenopila, Alveolata and Rhizaria) and, Haptista. (lines 103-105) Within the TSAR supergroup, some MIPs cluster with the families PIP, GIP, and MIPE, whereas other MIPs cluster in a new family specific to TSAR organisms, named Large Intrinsic Proteins (LIPs).
7	Introduction, line 54: the word "sanitary" refers to diseases caused by hygiene conditions, which is not the case of trypanosomatids of medical importance; they cause vector-borne diseases.	We thank the referee for the comment. We corrected the phrase: Introduction (lines 56-58) Before: The Trypanosomatida order of kinetoplastids (Euglenozoa, Discoba) gathers a vast diversity of parasitic protozoans that cause significant worldwide sanitary problems infecting humans and livestock. Now: The Trypanosomatida order of kinetoplastids (Euglenozoa, Discoba) gathers a vast diversity of parasitic protozoans that cause significant worldwide health problems infecting humans and livestock.
8	It is interesting that T. brucei lost AQPXs and retained GLPs, whereas T. cruzi lost GLPs and kept AQPXs. Are there other similar examples of unusual gene family loss/retention in these parasites? It would enrich the manuscript if the authors would speculate potential reasons or discuss similar examples. The discussion seems overall quite focused on	There are examples in the literature of proteins that are exclusively present in T. brucei or T. cruzi (the most studied species among African and American trypanosomes, respectively) and their closely related Leishmania spp. Comparative studies of the three genomes revealed insertions or substitutions were the processes involved in the acquisition of the lineage-specific genes found in L. major, T. brucei and T. cruzi genomes (El-Sayed et al., 2005). These trypanosomatid species genomes revealed large syntenic regions interspersed by clusters of species-specific multigene families related to the parasitic lifestyle. Examples of these protein families are mucins and trans sialidases in T. cruzi, variant surface glycoproteins in T. brucei and δ-amastin, cysteine peptidases and surface antigen proteins in L. major (El-Sayed et al., 2005; Jackson, 2015; Reis-cunha et al., 2018). Species-specific genes are physiologically relevant since they are members of surface antigen families and thus relate to host-parasite interactions and parasite survival (El-Sayed et al., 2005). Therefore, these lineage-specific genes do not necessarily correspond to homologous genes coding for proteins of the same superfamily. Regarding the MIP superfamily asymmetry, the scenario is different. Though African and American trypanosomes hold specific MIP types (GLPs or AQPXs), all isoforms belong to the same superfamily as opposed to species-specific

trypanosomes and lacks some broad discussions of this type of events.	surface proteins. Moreover, all MIP genes localize in syntenic regions of the genomes of the subfamily Trypanosomatinae. Studies that focused on the emergence of parasitism in trypanosomatid species reported examples of gene losses occurring independently in Trypanosoma and Leishmania genera (i.e. Cathepsins), revealing a genomic reduction through parasite diversification (after free living B. saltans branched) that often resulted in the asymmetric variety of ancestral gene repertoires among the trypanosomatid lineages (Jackson et al., 2016). There are also examples of inclusive (occurring in all species) and exclusive (placed after diversification) gene duplications in trypanosomatid lineages (i.e. amino acid transporters, nucleoside transporters and amastins) (Jackson et al., 2016). So, lineage specific gene losses and exclusive expansions are common events occurring in the evolutionary history of extant trypanosomatid species. Nevertheless, each example is unique and therefore we did not address a broader discussion but focused on depicting the most parsimonious scenario for each MIP family gain and loss among kinetoplastid extant species. Regarding the MIP superfamily, we do not have evidence to speculate on the possible reasons for the asymmetry found among African and American trypanosomes. Given their lifestyles are utterly different, unraveling the physiological role of these proteins in each one will aid understand this asymmetry. Interestingly, among other unicellular organisms, and particularly within Stramenopiles (TSAR eukaryotic supergroup), Oomycetes have several GLPs but no AQP while other groups (Bacillariophyceae, Phaophyceae and Pelagophyceae) have less isoforms but AQP and GLP both present. In Abascal et al. (2014), authors refer to this pattern as evidence of an evolutionary relationship between the loss of AQPs and consequent expansion of GLP (or the other way around). In kinetoplastids, expansion of AQPXs or GLPs also takes place in the absence of the other type of MIP (i.e in the metakinetoplastid ancestor AQPX expanded prior to GLP gain in a trypanosomatid ancestor and in the T. brucei and T. evansi ancestor, expansion of GLP took place after AQPX were lost). We modified the manuscript by adding a new paragraph addressing MIP asymmetry among trypanosomatids, as suggested by the referee. This modification follows the detailed description of kinetoplastid MIPs gains and losses compatible with our phylogenetic and syntenic data under the subtitle Gains and losses of MIPs in Trypanosome genomes Results and discussion (addition in lines 395-407: new, independent paragraph): So, genera or species-specific gene gains and losses resulted in an asymmetric repertoire of MIPs in extant trypanosomatid parasites. Such processes are usual in the evolutionary history of other protein families among T. brucei, T. cruzi and Leishmania species (i.e. cathepsins, amastins, nucleoside and amino acid transporters) (Jackson, 2010; Jackson et al., 2016). Utterly different lifestyles and hosts might relate to species-specific gene expansions and losses. For example, amastin diversity remained unchanged until the origin of Leishmania. So, the specific δ-amastin expansion that occurred in this species was speculated to relate to Leishmania's vertebrate parasitism given the absence of this gene family in related monoxenous species (insect-restricted parasitism) (Jackson,
--	--

2010). Regarding the MIP superfamily, biological relevance of each family (GLP and AQPX) in trypanosomes still remains obscure though the asymmetric pattern is coherent with the proposal of an evolutionary relationship between the loss of AQPs and consequent expansion of GLP (or the other way around) based on observations of other unicellular organisms like Oomycetes, that hold numerous GLP isoforms and none AQPs (Abascal et al., 2014).

Revision Table 1. SRAs of RNA-seq experiment of *P. confusum* blasted against the genomes of *P. confusum* and *L. pyrrhocris*.

SRA-Query	Genome-Subject	N hits	Total hits/SRA	Relative Hits %	BUSCO
SRR6231225	PconfusumCUL13	11690	11749	99.49	94.6
	LpyrrhocrisH10	59		0.502	
SRR6231232	PconfusumCUL13	10451	10464	99.87	94.6
	LpyrrhocrisH10	13		0.124	
SRR6231231	PconfusumCUL13	10705	10720	99.86	94.6
	LpyrrhocrisH10	15		0.139	
SRR6231224	PconfusumCUL13	12313	12488	98.59	93.1
	LpyrrhocrisH10	175		1.401	
SRR6231227	PconfusumCUL13	54215	62729	86.42	3.8
	LpyrrhocrisH10	8514		13.57	
SRR6231228	PconfusumCUL13	12683	23123	54.85	90.80
	LpyrrhocrisH10	10440		45.14	
SRR6231226	PconfusumCUL13	11549	11603	99.53	94.6
	LpyrrhocrisH10	54		0.465	
SRR6231223	PconfusumCUL13	10661	10674	99.87	54.5
	LpyrrhocrisH10	13		0.121	

N°	Referee #3's comments	Author's reply
	The ms by Tesan et al. describes a theoretical analysis of a large number of aquaporin (AQP) protein sequences of the kinetoplastids using sequence similarity networks (SSN) and phylogenetic trees. As the main result the authors combine the previously termed AQP alpha/beta/gamma/delta types of sequences to a cluster, which they term AQPX. Aquaporins are channels for water and small uncharged solutes, mainly glycerol and urea. However, other small neutral molecules can pass, such as ammonia, hydrogen peroxide, or even protonated short-chain monocarboxylic acids. The variety is large due to limited selectivity capacities of the AQPs that is based alone on charge exclusion and size selection in the selectivity filter region. From an evolutionary point of view the provided analysis may be of interest, namely the notion that African and American trypanosomes express a shifted profile of AQPs. In terms of relevance for drug treatment as suggested by the authors, this probably does not hold true, though. In this regard and concerning selectivity mechanisms, the ms contains some errors, and outdated or one-sided views, which should be corrected.	
0	From an evolutionary point of view the provided analysis may be of interest, namely the notion that African and American trypanosomes express a shifted profile of AQPs. In terms of relevance for drug treatment as suggested by the authors, this probably does not hold true, though.	We appreciate the comment on the interest of our work from an evolutionary point of view. Indeed, our main results relate to comprehensive phylogenetic, syntenic, and primary sequence data analyses and our statements on drug treatment were intended to contextualize our findings and mention future perspectives. Throughout the manuscript we mentioned the role of MIPs as drug targets as potential, possible and further to be addressed. Moreover, our results show that AQPXs are utterly different from MIPs involved in parasite's sensitivity/resistance to drug treatment (TbAQP2, LmAQP1). Therefore, we do not have evidence yet to place them as possibly involved in drug internalization or permeability and thus this statement was not included in any section of our manuscript. Nonetheless, we find the description of the key MIP residues, together with our thorough phylogenetic analysis, are useful to further hypothesize which solutes (therapeutic or not) can pass through AQPXs and eventually evaluate their potentiality as drug targets or gateways.
1	137, and others: The wording "transport" must not be used in connection with channel functionality. Please correct throughout the ms.	Despite MIPs being channels and not transporters, it is frequent to find the word "transport" in the literature. Most of the published works on MIPs use the word "transport" to refer to permeability to water and other solutes. Thus, functionality of MIP channels is usually denoted as "transport" and this word was used in the initial groundbreaking works, P. Agre's lectures (Verkman and Mitra 2000; Agre et al., 1993; Agre, 2006) and in the most updated reviews and research papers on the subject (Laloux et al., 2018; Abascal et al., 2014; Finn et al., 2014; Pommerrenig et al., 2020; Tyerman et al., 2021). However, as happens with the term "function" whose meaning is not straightforward (Thomas, 2017), we agree that using "transport" can be problematic if a wide perspective on function is not considered since this term does not directly denote "channel functionality". Also, if the reader is not a specialist in MIPs can be confused when the term transport is used. So, to avoid possible misinterpretation we followed reviewer suggestions and modified the manuscript consequently. We replaced "transport" with "permeation", "facilitated diffusion" and "passing through".

141, “previously undescribed family”: the authors give a literature reference to this type of AQP isoforms, hence it has been described. 2	As stated in the Results and discussion section (lines 173-174), our study showed that AqpXs integrate a well-supported grade among prokaryotic MIPs. Therefore, this is evidence of AQPX being originated before the emergence of the Eukarya domain of life. As far as we know, this is the first phylogenetic study that provides evidence of the evolutionary origin of this group of proteins. Altogether, our results support this group being a MIP family and, as a family, it has not been previously described. Notwithstanding, as the referee mentioned, we are aware of the characterization results reported and reviewed for some eukaryotic isoforms of AQPXs - T. cruzi α and L. major α-δ paralogues (Montalvetti et al., 2004; Biyani et al., 2011; Neumann et al., 2020)- but in none of them were these MIPs described as belonging to a specific AQP family, but water-selective aquaporins or TIP-like aquaporins in accordance to previously published phylogenetic results (Abascal et al., 2014; Beitz, 2005; Quintana and Field, 2021). We mentioned this issue in lines 108-110 of the introduction where we reviewed all published literature on Eukaryotic MIP phylogeny. The complete paragraph mentioned by the referee states: “Here we place trypanosomatid channels in the sequence-function space of the large MIP superfamily and expose their presence in a previously undescribed family, here named aquaporin X (AQPX).” Our intention in this phrase was to point to the novelty of the place occupied by trypanosomatid MIPs in the SSN, not to point to the functional description of trypanosomatid MIP isoforms. To avoid confusing statements, we explain more in detail what is new and modified other phrases of the manuscript: Abstract (lines 40-44) Before: Here we place trypanosomatid channels in the sequence-function space of the large MIP superfamily and expose their presence in a previously undescribed family, here named aquaporin X (AQPX). Now: Here we placed trypanosomatid channels in the sequence-function space of the large MIP superfamily through a sequence similarity network. This analysis exposes that trypanosomatid aquaporins integrate a distant cluster from the currently defined MIP families, here named aquaporin X (AQPX). Introduction (lines 116-118) Before: In this work, we show that two MIP families expanded among trypanosomatids: GLP and a previously undescribed MIP family, named here AQPX. Now: In this work, we show that two MIP families expanded among trypanosomatids: GLP and a MIP family previously undescribed as such, named here AQPX. Results and Discussion (lines 504-507) Before: In conclusion, we depicted here the complex universe of MIPs through a SSN, clearly exposing that trypanosomatids carry GLPs and AQPXs, a newly described type of MIPs. AQPXs compose a cluster far away from the already characterized MIPs. Now: In conclusion, we depicted here the complex universe of MIPs through a SSN, clearly exposing that trypanosomatids carry GLPs and AQPXs. AQPXs compose a
--	--

		cluster far away from the already characterized MIPs and, our phylogenetic studies support that they integrate a newly defined MIP family.
3	168: The functional role of diverse NPA motifs is overrated throughout the ms, probably because statements are based on older references. The main proton filter resides in the “selectivity filter” region, see Wu et al. EMBO J. 2008 and Kosinska Ericsson et al. Science 2013.	We have updated our references and modified the manuscript to avoid transmitting a wrong idea about the role of each MIP constriction. Introduction (lines 69-76) Before: Two NPA (Asn-Pro- Ala) motifs in the middle part of the pore, regulate water conductance and operate as a proton-excluding filter. MIPs also have a selectivity filter (SF), known as aromatic/Arginine (ar/R), which executes a primary transport role. The residues of this filter are related to the functional properties of the channel and, interestingly, play a central role in trypanosomatid drug transport, i.e., their mutation may lead to drug resistance events. Now: Two NPA (Asn-Pro-Ala) motifs in the middle part of the pore regulate water conductance, operate as a barrier for the passage of inorganic cations (such as Na⁺ and K⁺) (Wu et al., 2009; Wree et al., 2011), and also participate in proton filtration (Murata et al., 2000; Tajkhorshid et al., 2002). Still, protons are fully blocked at the selectivity filter (SF) (Wu et al., 2009; Wree et al., 2011; Eriksson et al., 2013), known as aromatic/Arginine (ar/R), which also executes a primary permeation role. The residues of this filter are related to the functional properties of the channel (De Groot and Grubmüller, 2005; Hub and de Groot, 2008) and, interestingly, play a central role in trypanosomatid drug uptake, i.e., their mutation may lead to drug resistance events.
4	1387: variations in the NPA motifs at the second and third position, e.g. NSA/NPS, have never been shown to have consequences on the selectivity. The relevant position is the “N”.	To our knowledge, no functional studies are reporting the participation of the second or third position of the NPA motifs of TbAQP2 in the solute selectivity of this MIP. But it is important to mention that mutants of the NSA/NPS motif to our knowledge have not been analyzed regarding the transport of glycerol, water, or other “classical” MIP solutes and therefore the possibility cannot be excluded. Evidence from other MIPs points to the necessity of testing this hypothesis. For example, mouse AQP11 holds an N-terminal NPA variation (NPC) that failed to permeate water in the NPA mutant (Ikeda et al., 2011). With this information and our results on the conservation of key MIP residues (within Trypanosomatids) we consider that it is necessary to report any NPA variations and their functional consequences when possible. We found that GLPs from the Trypanosomatinae subfamily had conserved crucial MIP residues and therefore we decided to highlight the only two isoforms with different NPA motifs and SF without speculating on their contribution in solute selectivity. We agree with the reviewer that the literature shows that N is the most relevant NPA position of MIPs (using other non trypanosomatid MIPs) and thus, we modify the sentence in I387 and added literature to support the statement. Results and Discussion (lines 433-436) Before: These AQP2 are the only GLPs with non-canonical NPA motifs (NSA and NPS). Now: These AQP2s are the only trypanosomatid GLPs with non-canonical NPA motifs

		(NSA and NPS). Importantly, the N in the first position of the motifs that have been proved to be important for cation blockage (Wu et al., 2009; Wree et al., 2011) is conserved in T. brucei and T. evansi AQP2.
5	1393: the matter whether pentamidine can actually pass TbAQP2 is not clear. An alternative model explains TbAQP2-dependent uptake of pentamidine by high-affinity binding to a uniquely exposed aspartate in the selectivity filter “D” (see point 6) and endocytosis of the complex, see Song et al. PLoS Pathog. 2016.	We agree with the reviewer that pentamidine uptake is still an unsettled matter. We included both hypotheses in our manuscript and added the paper mentioned by the referee (Song et al. 2016) as cited literature. Results and Discussion (lines 441-448) Before: TbAQP2 SF is wider and more aliphatic than others, and that contributes to pentamidine passing through. It is vital to bear that no other T. brucei MIP transports pentamidine (TbAQP1 nor TbAQP3), whereas all T. brucei MIPs transport water, glycerol, and metalloids in a similar way. Now: TbAQP2 SF is wider and more aliphatic than others. A first hypothesis sustains that this feature contributes to pentamidine passing through (Alghamdi et al., 2020; Quintana et al., 2020) and a second one that the unique SF in combination with a consequently exposed Asp (D265, Froger position P2), allows a high affinity binding of pentamidine followed by endocytosis (Song et al., 2016). It is vital to bear that no other T. brucei MIP participates in pentamidine uptake (TbAQP1 nor TbAQP3), whereas all T. brucei MIPs facilitate the diffusion of water, glycerol, and metalloids in a similar way.
6	1.427: The specialty of TbAQP2 is the “L” in the “R” position of the selectivity filter IN COMBINATION with keeping the “D” right next to the “R”. The named other AQPs lacking a positively charged residues in the selectivity filter also have a neutral residue in the following “D” position. This “D” or absence thereof should be integrated into assumptions on structure-function relationships.	Our residue analysis included those key MIP positions related to permeability and selectivity (NPA and SF) and also the Froger positions that were basically defined by observing the difference among MSA of GLPs and AQPs (Froger et al., 1998). The D named by the referee (D265 of TbAQP2) corresponds to the Froger position P2 of GLPs that is conserved among trypanosomes. Indeed, AQPXs lack a positively charged residue like D occupying Froger P2 position and hold a neutral residue (A) instead. Moreover, AQPX isoforms have conserved Froger positions (Figure 7 and lines 462-464) with AQP- like residues occupying them. In I. 427 (now 479) the SF of AQPXs is compared to the one carried by other AQPs. The phrase stands out the absence of both the aromatic amino acid and the R. Froger position P2 was not discussed here given that it is not part of the SF residues, it is usually occupied by S or A in AQPs and there are no available reports that we know of, on the impact of this residue regarding AQP’s selectivity. Regarding GLPs key MIP residues, we agree with the referee that in the light of the published results on pentamidine’s supposed binding as a consequence of the uniquely exposed D in TbAQP2 (Song et al., 2016) it is necessary to specifically refer in the text if AQPXs hold a similar residue in that position. That information was provided in Figure 7 but not discussed in the manuscript. So, taking all that in consideration we changed the manuscript and highlighted AQPXs do not hold a D right next to the R position of the SF (Froger position P2) but a neutral amino acid. Additionally, the relevance of the L in the SF’s R position combined with the D exposure in TbAQP2 was already addressed in response to comment #5. Results and Discussion

		(lines 462-464) Before: AQPXs display generally conserved Froger positions (AQYW from P2 to P5) (Fig. 7b). Now: AQPXs display generally conserved Froger positions (AQYW from P2 to P5) (Fig. 7b) with AQP- like residues occupying them. (lines 479-486) Before: Compared with SF in classical water channels AQP1-likes and PIPs, the SF of AQPXs do not keep the R in Loop E (LE2), nor the aromatic amino acids in TM2, having, instead, aliphatic residues (Fig. 7b). That may give place to more hydrophobic and broader filters. Many AQPXs (except the β orthologs) have a V in the LE2 position. Now: Compared with SF in classical water channels AQP1-likes and PIPs, the SF of AQPXs do not keep the R in Loop E (LE2), nor the aromatic amino acids in TM2, having, instead, aliphatic residues (Fig. 7b). That may give place to more hydrophobic and broader filters. Though their SF is aliphatic, they also hold an aliphatic uncharged residue (an A) where TbAQPs have an acidic amino acid (Froger position P2) and the impact of these differences and the eventual exposure of other AQPX residues affecting permeation or selectivity needs to be addressed by further structural and functional research. Finally, many AQPXs (except the β orthologs) have a V in the LE2 position.
--	--	---

❖ Additional modifications and author's comments

While revising our manuscript we found an error within a phrase in the results section and one in the references. Also, the original legend of Figure 5 was not clear enough. We added information for clarity purposes and interpretation of the figure is now easier. We apologize for these mistakes and assure the main results are not affected by them. We have now corrected as follows:

Results (lines 229-234)

Before: Inside the Euglenida group, there are only two phototrophic organisms with sequenced genomes (the marine *Eutreptiella gymnastica* and the freshwater *Euglena gracilis*) and a heterotrophic organism with a transcriptome available (*Rhabdomonas constata*).

Now: Inside the Euglenida group, only **one** phototrophic organism **(the freshwater *Euglena gracilis*) has a sequenced genome and transcriptome available and other two organisms (the phototrophic *Eutreptiella gymnastica* and the heterotrophic *Rhabdomonas constata*) have transcriptomes available.**

Legend Figure 5

Before:

Figure 5. Synteny analysis of Trypanosomatid MIPs. Comparative gene organization of the regions where a) AQPXs and b) GLPs are found. The analyzed regions comprise 10 Kb down and up-stream of each MIP. Homologous genes are vertically aligned. The graph shows when the syntenic genes are observed using both SimpleSynteny analysis and TritrypDB (black arrow box) or one of these methods (dark grey arrow box). Genes Absent in the reference genome (*T. cruzi* or *T. brucei*) were not detected by SimpleSynteny but were observed in TritrypDB (light grey arrow box).

Now:

Figure 5. Synteny analysis of Trypanosomatid MIPs. Comparative gene organization of the regions where a) AQPXs and b) GLPs are found. The analyzed regions comprise 10 Kb down and up-stream of each MIP in the reference genomes (*T. cruzi* or *T. brucei*) and the equivalent syntenic regions in the other genomes. For clarity, in this figure only the first 3 genes for down and up-stream regions are shown. Homologous genes are vertically aligned. The graph shows when the syntenic genes are observed using both SimpleSynteny analysis and TritrypDB (black arrow box) or one of these methods (dark grey arrow box). Genes Absent in the reference genome (*T. cruzi* or *T. brucei*) were not detected by SimpleSynteny but were observed in TritrypDB (light grey arrow box).

References

Before: Former reference 30, cited in former line 487: Butenko, A. et al. Reductionist Pathways for Parasitism in Euglenozoans? Expanded Datasets Provide New Insights. *Trends Parasitol.* 37, 100–116 (2021).

Now: Actual reference 34, cited in line 707: Butenko, A. et al. Evolution of metabolic capabilities and molecular features of diplomonads, kinetoplastids, and euglenids. *BMC Biol.* 18, 1–28 (2020).

❖ Bibliography used in this response to the referees

- Abascal Federico, Irisarri Iker, and Zardoya Rafael. 2014. “Diversity and Evolution Of membrane Intrinsic Proteins.” *Biochimica et Biophysica Acta Journal* 1840: 1468–81.
- Afgan Enis, Baker Dannon, Van Den Beek Marius, Bouvier Dave, Chilton John, Clements Dave, Coraor Nate, et al. 2018. “The Galaxy platform for accessible, reproducible and collaborative biomedical analyses: 2018 update.” *Nucleic Acids Research* 46 (May): 537–44. doi.org/10.1093/nar/gky379.
- Agre Peter, Preston G M, Smith B L, Jung J S, Raina S, Moon C, Guggino W B, and Nielsen S. 1993. “Aquaporin CHIP: the archetypal molecular water channel.” *The American Journal of Physiology* 265 (4 Pt 2): F463-76.
- Agre Peter. 2006. “The aquaporin water channels.” *Proceedings of the American Thoracic Society* 3 (1): 5–13. doi.org/10.1513/pats.200510-109JH.
- Alghamdi Ali H., Munday Jane C., Campagnaro Gustavo D., Gurvič Dominik, Svensson Fredrik, Okpara Chinyere E., Kumar Arvind, et al. 2020. “Positively selected modifications in the pore of TbAQP2 allow pentamidine to enter *Trypanosoma brucei*.” *ELife* 9: 1–33. doi.org/10.1101/2020.03.08.982751.
- Beitz Eric. 2005. “Aquaporins from pathogenic protozoan parasites: structure, function and potential for chemotherapy.” *Biology of the Cell* 97 (6): 373–83. doi.org/10.1042/BC20040095.
- Biyani Neha, Swati Mandal, Chandan Seth, Malika Saint, Krishnamurthy Natarajan, Indira Ghosh, and Rentala Madhubala. 2011. “Characterization of *Leishmania donovani* aquaporins shows presence of subcellular aquaporins similar to Tonoplast Intrinsic Proteins of plants.” *PLoS ONE* 6 (9). doi.org/10.1371/journal.pone.0024820.
- Butenko Anzhelika, Opperdoes Fred R., Flegontova Olga, Horák Aleš, Hampl Vladimír, Keeling Patrick, Gawryluk Ryan M.R., Tikhonenkov Denis, Flegontov Pavel, and Lukeš Julius. 2020. “Evolution of metabolic capabilities and molecular features of diplomonads, kinetoplastids, and euglenids.” *BMC Biology* 18 (1): 1–28. doi.org/10.1186/s12915-020-0754-1.
- Chrispeels Maarten J, and Agre Peter. 1994. “Aquaporins: water channel proteins of plant and animal cells.” *Trends in Biochemical Sciences* 19(10):421-5. doi: 10.1016/0968-0004(94)90091-4.
- Copp Janine N, Akiva Eyal, Babbitt Patricia C, and Tokuriki Nobuhiko. 2018. “Revealing unexplored sequence-function space using sequence similarity networks.” *Biochemistry* 57 (31), 4651-4662. doi.org/10.1021/acs.biochem.8b00473.
- Danecek Petr, Bonfield James K, Liddle Jennifer, Marshall John, Ohan Valeriu, Pollard Martin O, Whitwham Andrew, et al. 2021. “Twelve years of SAMtools and BCFtools,” *GigaScience* 10 (2), giab008.

doi.org/10.1093/gigascience/giab008.

- El-Sayed Najib M., Myler Peter J., Blandin Gaëlle, Berriman Matthew, Crabtree Jonathan, Aggarwal Gautam, Caler Elisabet, et al. 2005. “Comparative genomics of trypanosomatid parasitic protozoa.” *Science* 309 (5733): 404–9. doi.org/10.1126/science.1112181.
- Eriksson Urszula Kosinska, Fischer Gerhard, Friemann Rosmarie, Enkavi Giray, Tajkhorshid Emad, and Neutze Richard. 2013. “Subangstrom resolution X-ray structure details aquaporin-water interactions.” *Science* 340 (6138): 1346–49. doi.org/10.1126/science.1234306.
- Finn Roderick Nigel, Chauvigné François, Hlidberg Jón Baldur, Cutler Christopher P., and Cerdà Joan. 2014. “The lineage-specific evolution of aquaporin gene clusters facilitated tetrapod terrestrial adaptation.” *PLoS ONE* 9 (11): 1–38. doi.org/10.1371/journal.pone.0113686.
- Froger A, Tallur B, Thomas D, and Delamarche C. 1998. “Prediction of functional residues in water channels and related proteins.” *Protein Science* 7 (6): 1458–68. doi.org/10.1002/pro.5560070623.
- Groot Bert L. De, and Grubmüller Helmut. 2005. “The dynamics and energetics of water permeation and proton exclusion in aquaporins.” *Current Opinion in Structural Biology* 15 (2): 176–83. doi.org/10.1016/j.sbi.2005.02.003.
- Gurevich Alexey, Saveliev Vladislav, Vyahhi Nikolay, and Tesler Glenn. 2013. “QUAST : Quality assessment tool for genome assemblies” *Bioinformatics* 29 (8): 1072–75. doi.org/10.1093/bioinformatics/btt086.
- Hub Jochen S, and de Groot Bert L. 2008. “Mechanism of selectivity in aquaporins and aquaglyceroporins.” *Proceedings of the National Academy of Sciences of the United States of America* 105 (4): 1198–1203. doi.org/10.1073/pnas.0707662104.
- Ikeda Masahiro, Andoo Ayaka, Shimono Mariko, Takamatsu Natsuko, Taki Asaka, Muta Kanako, Matsushita Wataru, et al. 2011. “The NPC motif of Aquaporin-11 , unlike the NPA motif of known aquaporins , is essential for full expression of molecular function.” *Journal of Biological Chemistry* 286 (5): 3342–50. doi.org/10.1074/jbc.M110.180968.
- Imhoff Johannes F, Rahn Tanja, Künzel Sven, and Neulinger Sven C. 2018. “Photosynthesis is widely distributed among proteobacteria as demonstrated by the phylogeny of PufLM reaction center proteins.” *Frontiers in Microbiology* 8: 2679. doi.org/10.3389/fmicb.2017.02679.
- Ishibashi Kenichi, Tanaka Yasuko, and Morishita Yoshiyuki. 2020. "Perspectives on the evolution of aquaporin superfamily". *Vitamins and Hormones* 112: 1-27. doi.org/10.1016/bs.vh.2019.08.001.
- Jackson Andrew P., Otto Thomas D., Aslett Martin, Armstrong Stuart D., Bringaud Frederic, Schlacht Alexander, Hartley Catherine, et al. 2016. “Kinoplastid phylogenomics reveals the evolutionary innovations associated with the origins of parasitism.” *Current Biology* 26 (2): 161–72. doi.org/10.1016/j.cub.2015.11.055.
- Jackson Andrew P. 2015. “Genome evolution in trypanosomatid parasites.” *Parasitology*, 142(S1), S40-S56. doi.org/10.1017/S0031182014000894.
- Jackson Andrew P. 2010. “The evolution of amastin surface glycoproteins in trypanosomatid parasites Research Article.” *Molecular Biology and Evolution* 27 (1): 33–45. doi.org/10.1093/molbev/msp214.
- Laloux Timothée, Junqueira Bruna, Maistriaux Laurie C., Ahmed Jahed, Jurkiewicz Agnieszka, and Chaumont François. 2018. “Plant and mammal aquaporins: same but different.” *International Journal of Molecular Sciences* 19 (2). doi.org/10.3390/ijms19020521.
- Langmead Ben, and Salzberg Steven L. 2012. “Fast gapped-read alignment with Bowtie 2.” *Nature Methods* 9 (4): 357–60. doi.org/10.1038/nmeth.1923.
- Minh Bui Quang, Anh Minh, Nguyen Thi, and Von Haeseler Arndt. 2013. “Ultrafast approximation for phylogenetic bootstrap” *Molecular Biology and Evolution* 30 (5): 1188–95. doi.org/10.1093/molbev/mst024.
- Montalvetti Andrea, Rohloff Peter, and Docampo Roberto. 2004. “A functional aquaporin co-localizes with the vacuolar proton pyrophosphatase to acidocalcisomes and the contractile vacuole complex of *Trypanosoma Cruzi*.” *Journal of Biological Chemistry* 279 (37): 38673–82. doi.org/10.1074/jbc.M406304200.
- Murata Kazuyoshi, Mitsuoka Kaoru, Hirai Teruhisa, Walz Thomas, Agre Peter, Heymann J Bernard, Engel Andreas, and Fujiyoshi Yoshinori. 2000. “Structural determinants of water permeation through aquaporin-1.” *Nature* 407 (6804): 599–605. doi.org/10.1038/35036519.
- Neumann Lucas S.M., Dias Artur H.S., and Skaf Munir S.. 2020. “Molecular modeling of aquaporins from *Leishmania Major*.” *Journal of Physical Chemistry B* 124 (28): 5825–36.

doi.org/10.1021/acs.jpcc.0c03550.

- Nguyen Lam Tung, Schmidt Heiko A., Von Haeseler Arndt, and Minh Bui Quang. 2015. "IQ-TREE: A Fast and effective stochastic algorithm for estimating maximum-likelihood phylogenies." *Molecular Biology and Evolution* 32 (1): 268–74. doi.org/10.1093/molbev/msu300.
- Pao GM, Wu LF, Johnson KD, Höfte H, Chrispeels MJ, Sweet G, Sandal NN, Saier MH Jr. 1991. "Evolution of the MIP family of integral membrane transport proteins." *Molecular Microbiology* 5 (1): 33–37. doi.org/10.1111/j.1365-2958.1991.tb01823.x
- Perez Di Giorgio Juliana, Soto Gabriela, Alleva Karina, Jozefkowicz Cintia, Amodeo Gabriela, Muschiatti Jorge Prometeo, and Ayub Nicolás Daniel. 2014. "Prediction of aquaporin function by integrating evolutionary and functional analyses." *Journal of Membrane Biology*. doi.org/10.1007/s00232-013-9618-8.
- Pommerrenig Benjamin, Diehn Till A., Bernhardt Nadine, Bienert Manuela D., Mitani-Ueno Namiki, Fuge Jacqueline, Bieber Annett, et al. 2020. "Functional evolution of Nodulin 26-like Intrinsic proteins: from bacterial arsenic detoxification to plant nutrient transport." *New Phytologist* 225 (3): 1383–96. doi.org/10.1111/nph.16217.
- Quintana Juan F., Bueren-Calabuigid Juan, Zuccotto Fabio, de Koning Harry P., Horn David, and Field Mark C.. 2020. "Instability of Aquaglyceroporin (Aqp) 2 contributes to drug resistance in Trypanosoma Brucei." *PLoS Neglected Tropical Diseases* 14 (7): 1–26. doi.org/10.1371/journal.pntd.0008458.
- Quintana Juan F., and Field Mark C.. 2021. "Evolution, function and roles in drug sensitivity of trypanosome aquaglyceroporins." *Parasitology*, 8–13. doi.org/10.1017/S0031182021000354.
- Reis-cunha João Luís, Valdivia Hugo O, and Bartholomeu Daniella Castanheira. 2018. "Gene and chromosomal copy number variations as an adaptive mechanism towards a parasitic lifestyle in trypanosomatids," *Current Genomics* 19 (2): 87–97. doi.org/10.2174/1389202918666170911161311.
- Reizer Jonathan, Reizer Aiala, and Saier Milton H. 1993. "The MIP family of integral membrane channel proteins : sequence comparisons, evolutionary relationships, reconstructed pathway of evolution, and proposed functional differentiation of the two repeated halves of the proteins." *Critical Reviews in Biochemistry and Molecular Biology* 28 (3): 235–57. doi.org/10.3109/10409239309086796.
- Simão Felipe A., Waterhouse Robert M., Ioannidis Panagiotis, Kriventseva Evgenia V., and Zdobnov Evgeny M.. 2015. "BUSCO: Assessing Genome Assembly and Annotation Completeness with Single-Copy Orthologs." *Bioinformatics* 31 (19): 3210–12. doi.org/10.1093/bioinformatics/btv351.
- Skalický Tomáš, Dobáková Eva, Wheeler Richard J, Tesarová Martina, Flegontov Pavel, Jirsová Dagmar, Votýpka Jan, Yurchenko Vyacheslav, Ayala Francisco J, and Lukeš Julius. 2017. "Extensive flagellar remodeling during the complex life cycle of Paratrypanosoma , an early-branching trypanosomatid." *Proceedings of the National Academy of Sciences* 114 (44): 11757–62. doi.org/10.1073/pnas.1712311114.
- Song Jie, Baker Nicola, Rothert Monja, Henke Björn, Jeacock Laura, Horn David, and Beitz Eric. 2016. "Pentamidine is not a permeant but a nanomolar inhibitor of the Trypanosoma Brucei Aquaglyceroporin-2." *PLoS Pathogens* 12 (2): 1–14. doi.org/10.1371/journal.ppat.1005436.
- Souza Denise Andréa Silva De, Parada Pavoni Daniela, Krieger Marco Aurélio, and Ludwig Adriana. 2018. "Evolutionary analyses of myosin genes in trypanosomatids show a history of expansion, secondary losses and neofunctionalization." *Scientific Reports* 8 (1): 1–11. doi.org/10.1038/s41598-017-18865-y.
- Tajkhorshid Emad, Nollert Peter, Jensen Morten, Miercke Larry J.W., O'Connell Joseph, Stroud Robert M., and Schulten Klaus. 2002. "Control of the selectivity of the aquaporin water channel family by global orientational tuning." *Science* 296 (5567): 525–30. doi.org/10.1126/science.1067778.
- Tanifuji Goro, Cenci Ugo, Moog Daniel, Dean Samuel, Nakayama Takuro, Colp Morgan, Flegontov Pavel, Johnson-MacKinnon Jessica, and McPhee Michael. 2017. "Genome sequencing reveals metabolic and cellular interdependence in an amoeba- kinetoplastid symbiosis," *Scientific Reports* 7, 11688. doi.org/10.1038/s41598-017-11866-x.
- Thomas Paul D.. 2017. "The gene ontology and the meaning of biological function." In: Dessimoz C., Škunca N. (eds) *The Gene Ontology Handbook. Methods in Molecular Biology*, 1446. Humana Press, NY. doi.org/10.1007/978-1-4939-3743-1_2.
- Tyerman Stephen D, Mcgaughey Samantha A, Qiu Jiaen, Yool Andrea J, and Byrt Caitlin S. 2021. "Adaptable and multifunctional ion-conducting aquaporins." *Annual Review of Plant Biology* 72:1. doi.org/10.1146/annurev-arplant-081720-013608.
- Valach Matus, Moreira Sandrine, Hoffmann Steve, and Stadler Peter F. 2017. "Keeping it complicated :

- Mitochondrial genome plasticity across Diplonemids.” *Scientific Reports* 7, 14166. doi.org/10.1038/s41598-017-14286-z.
- Verkman AS, and Mitra Alok K. 2000. “Structure and function of aquaporin water channels.” *American Journal of Physiology-Renal Physiology* 278: F13–28. doi.org/10.1152/ajprenal.2000.278.1.F13.
- Viborg Alexander Holm, Terrapon Nicolas, Lombard Vincent, Gurvan Michel, Czjzek Mirjam, Henrissat Bernard, and Brumer Harry. 2019. “A subfamily roadmap of the evolutionarily diverse Glycoside Hydrolase family 16 (GH16).” *Journal of Biological Chemistry* 294 (44): 15973–86. doi.org/10.1074/jbc.RA119.010619.
- Weatherly Brent D., Boehlke Courtney, and Tarleton Rick L.. 2009. “Chromosome level assembly of the hybrid Trypanosoma Cruzi genome.” *BMC Genomics* 10: 1–13. doi.org/10.1186/1471-2164-10-255.
- Wree Dorothea, Wu Binghua, Zeuthen Thomas, and Beitz Eric. 2011. “Requirement for asparagine in the aquaporin NPA sequence signature motifs for cation exclusion.” *FEBS Journal* 278 (5): 740–48. doi.org/10.1111/j.1742-4658.2010.07993.x.
- Wu Binghua, Steinbronn Christina, Alsterfjord Magnus, Zeuthen Thomas, and Beitz Eric. 2009. “Concerted action of two cation filters in the aquaporin water channel.” *EMBO Journal* 28 (15): 2188–94. doi.org/10.1038/emboj.2009.182.
- Yazaki Euki, Ishikawa Sohta A., Kume Keitaro, Kumagai Akira, Kamaishi Takashi, Tanifuji Goro, Hashimoto Tetsuo, and Inagaki Yuji. 2017. “Global Kinetoplastea phylogeny inferred from a large-scale multigene alignment including parasitic species for better understanding transitions from a free-living to a parasitic lifestyle.” *Genes and Genetic Systems* 92 (1): 35–42. doi.org/10.1266/ggs.16-00056.
- Zardoya Rafael. 2005. “Phylogeny and evolution of the Major Intrinsic Protein family.” *Biology of the Cell* 97 (6): 397–414. doi.org/10.1042/bc20040134.
- Zardoya Rafael, Villalba Soraya. 2001. “A phylogenetic framework for the aquaporin family in eukaryotes.” *Journal of Molecular Evolution* 52: 391–404. doi.org/10.1007/s002390010169.

Reviewers' Comments:

Reviewer #1:

Remarks to the Author:

I think the authors have done a good job at addressing the comments. I feel they have tried a bit too hard to avoid much additional tree building, but I very much doubt that the results here would be overturned. I am happy to recommend acceptance.

Reviewer #2:

Remarks to the Author:

I apologize for the delayed response. The authors did an excellent job in addressing my comments. Added additional analysis, and responses were very comprehensive.

Reviewer #3:

None

Communications Biology - Response to reviewers

Article Title: **AQPX-cluster aquaporins and aquaglyceroporins are asymmetrically distributed in trypanosomes**

Authors: Fiorella Carla Tesan, Juan Ramiro Lorenzo Lopez, Karina Alleva, Ana Romina Fox.

❖ **A point-by-point response to the referee's comments is shown below.**

Referee	Referee's comments	Author's reply
#1	I think the authors have done a good job at addressing the comments. I feel they have tried a bit too hard to avoid much additional tree building, but I very much doubt that the results here would be overturned. I am happy to recommend acceptance.	We appreciate the comment and the recommendation.
#2	I apologize for the delayed response. The authors did an excellent job in addressing my comments. Added additional analysis, and responses were very comprehensive.	We appreciate the constructive comments. Additional analysis driven by the referee's suggestions and questions improved and provided strength/solidity to our work.

❖ **Additional modifications and author's comments**

While revising our manuscript we found minor errors that we have now corrected as follows (in red):

Abstract

- Together, our results expose the diversity of trypanosomatid MIPs and will aid further functional, structural, and physiological research needed to face the potentiality of the AQPXs as ~~trypanocidal gateways~~ gateways for trypanocidal drugs.

Results and Discussion

- Among trypanosomatids, α - δ AQPXs seem to have been lost two times in different branches of ~~parasite evolution~~ the evolutionary tree (in African trypanosomes and *B. ayalai*, Fig. 6a).
- Regarding the MIP superfamily, biological relevance of each family (GLP and AQPX) in trypanosomes still remains obscure though the asymmetric pattern is coherent with the proposal of an evolutionary relationship between the loss of AQPs and consequent expansion of GLPs (or the other way around) based on observations of other unicellular organisms like Oomycetes, that hold numerous GLP isoforms and none AQPs.
- Paragraphs split was modified as follows:

Before: The most recently acquired GLP of *T. brucei* and *T. evansi* (AQP2) present utterly divergent key MIP residues from the other GLPs (Fig. 7a). These AQP2s are the only GLPs with non-canonical NPA motifs (NSA and NPS). Importantly, the N in the first position of the motifs that have been proved to be important for cation blockage^{11,12} is conserved in *T. brucei* and *T. evansi* AQP2.

Interestingly, functional consequences of the absence of both classical NPA motifs in *TbAQP2* are related to pentamidine sensitivity since the restitution of the NPA-NPA blocked the uptake of the drug⁷. Regarding the selectivity filter, these AQP2 carry a rare signature (IVLL), which is drastically different from the fully conserved selectivity filter of other trypanosomatids...

After: The most recently acquired GLP of *T. brucei* and *T. evansi* (AQP2) present utterly divergent key MIP residues from the other GLPs (Fig. 7a). These AQP2s are the only GLPs with non-canonical NPA motifs (NSA and NPS). Importantly, the N in the first position of the motifs that have been proved to be important for cation blockage^{11,12} is conserved in *T. brucei* and *T. evansi* AQP2. Interestingly, functional consequences of the absence of both classical NPA motifs in *TbAQP2* are related to pentamidine sensitivity since the restitution of the NPA-NPA blocked the uptake of the drug⁷.

Regarding the selectivity filter, these AQP2 carry a rare signature (IVLL), which is drastically different from the fully conserved selectivity filter of other trypanosomatids...

Figure caption:

- In Figure 5 a phrase was repeated twice:

Figure 5. Synteny analysis of Trypanosomatid MIPs. Comparative gene organization of the regions where **a** AQPXs and **b** GLPs are found. The analyzed regions comprise 10 Kb down and up-stream of each MIP in the reference genomes (*T. cruzi* or *T. brucei*) and the equivalent syntenic regions in the other genomes. For clarity, in this figure only the first three genes for down and up-stream regions are shown. ~~The analyzed regions~~

~~comprise 10 Kb down and up stream of each MIP.~~ Homologous genes are vertically aligned. The graph shows when the syntenic genes are observed using both SimpleSynteny analysis and TritrypDB (black arrow box) or one of these methods (dark grey arrow box). Genes absent in the reference genome (*T. cruzi* or *T. brucei*) were not detected by SimpleSynteny but were observed in TritrypDB (light grey arrow box).

- In Figure 7 we corrected a grammatical error

Figure 7. Key MIP residues from GLP and AQPX subfamilies. Froger positions, NPA motifs and Selectivity filter residues of Discoba **a** GLPs and **b-c** AQPXs. The most frequent amino acid of a particular position in each clade is shown in bold first and then with dots. **b** Collapsed Discoba AQPXs phylogenetic tree reconstructed by maximum likelihood. **c** ~~Analysis of residues-analysis~~